# Vector-valued Gaussian Processes on Riemannian Manifolds via Gauge Independant Projected Kernels

**Michael Hutchinson**[*1]  **Alexander Terenin**[*2, 3]  **Viacheslav Borovitskiy**[*4]
**So Takao**[*5]  **Yee Whye Teh**[1]  **Marc Peter Deisenroth**[5]

[1]University of Oxford  [2]University of Cambridge  [3]Imperial College London
[4]St. Petersburg State University  [5]Centre for Artificial Intelligence, University College London

## Abstract

Gaussian processes are machine learning models capable of learning unknown functions in a way that represents uncertainty, thereby facilitating construction of optimal decision-making systems. Motivated by a desire to deploy Gaussian processes in novel areas of science, a rapidly-growing line of research has focused on constructively extending these models to handle non-Euclidean domains, including Riemannian manifolds, such as spheres and tori. We propose techniques that generalize this class to model vector fields on Riemannian manifolds, which are important in a number of application areas in the physical sciences. To do so, we present a general recipe for constructing gauge independent kernels, which induce Gaussian vector fields, i.e. vector-valued Gaussian processes coherent with geometry, from scalar-valued Riemannian kernels. We extend standard Gaussian process training methods, such as variational inference, to this setting. This enables vector-valued Gaussian processes on Riemannian manifolds to be trained using standard methods and makes them accessible to machine learning practitioners.

## 1 Introduction

Gaussian processes are an effective model class for learning unknown functions. They are particularly attractive for use within data-efficient decision systems, including Bayesian optimization [3, 32, 39], model-based reinforcement learning [34, 7], and active learning [21]. In these settings, Gaussian processes can represent and propagate uncertainty, as well as encode inductive biases as prior information in order to drive data efficiency. A key aspect of prior information is the geometry of the domain on which the Gaussian process is defined, which often encodes key properties, such as symmetry. Following the growing deployment of Gaussian processes, a number of recent works have focused on how to define Gaussian processes on non-Euclidean domains in ways that reflect their geometric structure [2, 1].

In many applications, such as climate science, quantities of interest are vector-valued. For example, global wind velocity modeling must take into account both speed and direction, and is represented by a vector field. On geometric domains, the mathematical properties of vector fields can differ noticeably from their Euclidean counterparts: for instance, one can prove that every smooth vector field on a sphere must vanish in at least one point [26]. Behavior such as this simultaneously highlights the need to represent geometry correctly when modeling vector-valued data, and presents a number of non-trivial technical challenges in constructing models that are mathematically sound.

---

[*]Equal contribution. Code: https://github.com/MJHutchinson/ExtrinsicGaugeIndependantVectorGPs.
For a general implementation, see https://github.com/GPflow/GeometricKernels/.

In particular, even the classical definition of a vector-valued Gaussian process—that is, a random function with multivariate Gaussian marginals at any finite set of points—already fails to be a fully satisfactory notion when considering smooth vector fields on a sphere. This is because tangent vectors at distinct points live within different tangent spaces, and it is not clear how to construct a cross-covariance between them that does not depend on a completely arbitrary choice of basis vectors within each space. Constructions that are independent of this choice of basis are called *gauge independent*, and recent work [45, 5, 15] in geometric machine learning has focused on satisfying this key property for convolutional neural networks that deal with non-Euclidean data.

Our contributions include the following. We (a) present a differential-geometric formalism for defining Gaussian vector fields on manifolds in a coordinate-free way, suitable for Gaussian process practitioners with minimal familiarity with differential geometry, (b) present a universal and fully constructive technique for defining prior Gaussian vector fields on Riemannian manifolds, which we term the *projected kernel* construction, and (c) discuss how to adapt key components in the computational Gaussian process toolkit, such as inducing point methods, to the vector field setting.

The structure of the paper is as follows. In Section 2, we define vector-valued Gaussian processes on smooth manifolds. We start by reviewing the multi-output Gaussian process set-up, which is typically used in machine learning. We then detail a differential-geometric formalism for defining vector-valued Gaussian processes on smooth manifolds. In Section 3, we provide a concrete construction for these Gaussian processes on Riemannian manifolds and discuss how they can be trained using variational sparse approximations. Section 4 showcases Gaussian vector fields on two tasks, namely weather imputation from satellite observations and learning the dynamics of a mechanical system.

## 2 Vector-valued Gaussian Processes on Smooth Manifolds

A vector-valued Gaussian process (GP) is a random function $\boldsymbol{f} : X \to \mathbb{R}^d$ such that, for any finite set of points $\boldsymbol{x} \in X^n$, the random variable $\boldsymbol{f}(\boldsymbol{x}) \in \mathbb{R}^{n \times d}$ is jointly Gaussian. Every such GP is characterized by its mean function $\boldsymbol{\mu} : X \to \mathbb{R}^d$ and matrix-valued covariance kernel $k : X \times X \to \mathbb{R}^{d \times d}$, which is a positive-definite function in the matrix sense. These functions satisfy $\mathbb{E}(\boldsymbol{f}(\boldsymbol{x})) = \boldsymbol{\mu}(\boldsymbol{x})$ and $\mathrm{Cov}(\boldsymbol{f}(\boldsymbol{x}), \boldsymbol{f}(\boldsymbol{x}')) = k(\boldsymbol{x}, \boldsymbol{x}')$ for any $\boldsymbol{x}, \boldsymbol{x}' \in X$. Here, dependence between function values is encoded in the kernel's variability along its input domain, and correlations between different dimensions of the vector-valued output are encoded in the matrix that the kernel outputs.

Consider a function $\boldsymbol{f}$ with $\boldsymbol{y} = \boldsymbol{f}(\boldsymbol{x}) + \boldsymbol{\varepsilon}$, where $\boldsymbol{\varepsilon} \sim \mathrm{N}(\boldsymbol{0}, \sigma^2 \mathbf{I})$ and training data $(\boldsymbol{x}, \boldsymbol{y})$. Placing a GP prior $\boldsymbol{f} \sim \mathrm{GP}(0, k)$ on the unknown function results in a GP posterior, whose mean and covariance are given by

$$\mathbb{E}(\boldsymbol{f} \mid \boldsymbol{y}) = \mathbf{K}_{(\cdot)\boldsymbol{x}}(\mathbf{K}_{\boldsymbol{x}\boldsymbol{x}} + \sigma^2 \mathbf{I})^{-1}\boldsymbol{y} \quad \mathrm{Cov}(\boldsymbol{f} \mid \boldsymbol{y}) = \mathbf{K}_{(\cdot,\cdot)} - \mathbf{K}_{(\cdot)\boldsymbol{x}}(\mathbf{K}_{\boldsymbol{x}\boldsymbol{x}} + \sigma^2 \mathbf{I})^{-1}\mathbf{K}_{\boldsymbol{x}(\cdot)}. \quad (1)$$

Here, $(\cdot)$ denotes an arbitrary set of test locations, $\mathbf{K}_{\boldsymbol{x}\boldsymbol{x}} = k(\boldsymbol{x}, \boldsymbol{x})$ is the kernel matrix, and $\mathbf{K}_{(\cdot)\boldsymbol{x}} = k(\boldsymbol{x}, \cdot)$ is the cross-covariance matrix between function values evaluated at the training and test inputs. The GP posterior can also be written as

$$(\boldsymbol{f} \mid \boldsymbol{y})(\cdot) = \boldsymbol{f}(\cdot) + \mathbf{K}_{(\cdot)\boldsymbol{x}}(\mathbf{K}_{\boldsymbol{x}\boldsymbol{x}} + \sigma^2 \mathbf{I})^{-1}(\boldsymbol{y} - \boldsymbol{f}(\boldsymbol{x}) - \boldsymbol{\varepsilon}), \qquad \boldsymbol{\varepsilon} \sim \mathrm{N}(\boldsymbol{0}, \sigma^2 \mathbf{I}) \qquad (2)$$

where $\boldsymbol{f}(\cdot)$ is the prior GP, and equality holds in distribution [47, 48]. These expressions form the foundation upon which Gaussian-process-based methods in machine learning are built.

Recent works have studied techniques for working with the expressions (1) and (2) when the input domain $X$ is a Riemannian manifold, focusing both on defining general classes of kernels [2], and on efficient computational techniques [47, 48]. In this setting, namely for $f : X \to \mathbb{R}$, defining kernels already presents technical challenges: the seemingly-obvious first choice one might consider, namely the geodesic squared exponential kernel, is ill-defined in general [13]. We build on these recent developments to model vector fields on manifolds using GPs. We do not consider manifold-valued generalizations of Gaussian processes, for instance $f : \mathbb{R} \to X$: various constructions in this setting are instead studied by Stroock [40], Émery [11], Mallasto and Feragen [29], and Mallasto et al. [30]. To begin, we review what a vector field on a manifold actually is.

### 2.1 Vector Fields on Manifolds

Let $X$ be a $d$-dimensional smooth manifold with $T_x X$ denoting its tangent space at $x$. Let $TX = \{(x, v) \mid x \in X, v \in T_x X\}$ be its *tangent bundle*, and let $T^*X = \{(x, \phi) \mid x \in X, \phi \in T_x^* X\}$

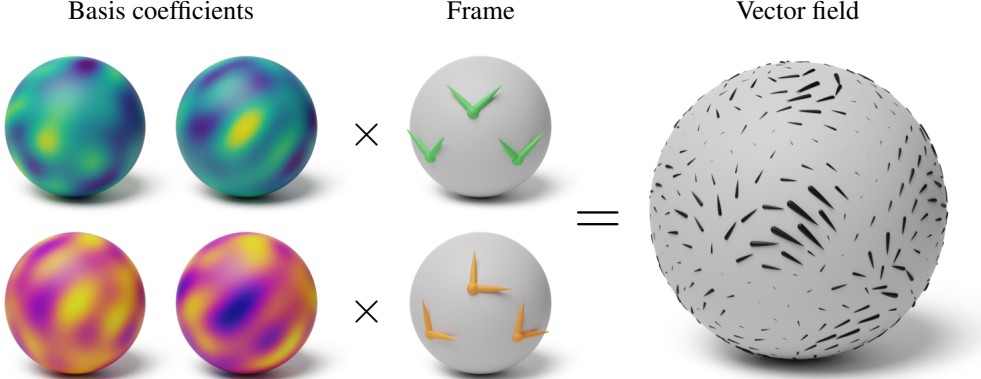

| Basis coefficients | Frame | Vector field |

Figure 1: Illustration on $\mathbb{S}^2$. Here we illustrate two possible bases (also called frames), consisting of green and orange basis vectors (center), that can be chosen locally on the manifold $\mathbb{S}^2$. The vector field on $\mathbb{S}^2$ (right) can be produced by taking two scalar fields (left) in each respective color, and combining them with the basis vectors (center) to form the vector field.

be its *cotangent bundle*—endow both spaces with the structure of smooth manifolds. Define the projection map $\text{proj}_X : TX \to X$ by $\text{proj}_X(x, v) = x$. A *vector field* on $X$ is a map that assigns each point in $X$ to a tangent vector attached to that point. More formally, a vector field is a *cross-section*, or simply a *section*, of the tangent bundle, which is a map $f : X \to TX$, such that $\text{proj}_X f(x) = x$ for all $x$.[1] A vector field is called *smooth* if this map $f$ is smooth.

To represent a vector field on a manifold numerically, one must choose a basis in each tangent space, which serves as a coordinate system for vectors in the tangent space. On many manifolds it is *impossible* to choose these basis vectors in a way that they vary smoothly in space.[2] This can be handled by working with local coordinates, or with bases that are non-smooth. Any chosen set of basis vectors is arbitrary, so objects constructed using them should not depend on this choice. Constructions that satisfy this property are called *gauge independent*. This notion is illustrated in Figure 1, and will play a key role in the sequel.

## 2.2 Gaussian Vector Fields

Upon reflecting on the above considerations in the context of GPs, the first issue one encounters is that, for a random vector field $f : X \to TX$, it is not clear what it means for finite-dimensional marginal distributions to be *multivariate Gaussian* given that $f$ takes its values in a bundle rather than a vector space. The first step towards constructing Gaussian vector fields, therefore, involves adapting the notion of *finite-dimensional marginal distributions* appropriately.

**Definition 1.** *Let $X$ be a smooth manifold. We say that a random vector field $f$ is* Gaussian *if for any finite set of locations $x_1, \ldots, x_n \in X$, the random vector $(f(x_1), \ldots, f(x_n)) \in T_{x_1}X \oplus \ldots \oplus T_{x_n}X$ is Gaussian (either in the sense of duality or in any basis: see Appendix A for details).*

This definition is near-identical to the Euclidean case: the only difference is that finite-dimensional marginals are now supported in a direct sum of tangent spaces, instead of $\mathbb{R}^{n \times d}$. With this definition, the standard multi-output GP properties, such as conditioning, carry over, virtually unmodified. Definition 1 is a natural choice: if we embed our manifold into Euclidean space, the induced GP is a vector-valued GP as defined in the beginning of Section 2.

---

[1] A vector field is *not the same as* a map $\tilde{f} : X \to \mathbb{R}^d$: an output value $f(x) \in TX$ formally consists of both a copy of the input point $x$, and vector within the tangent space $T_x X$ at this point. This encodes the geometric structure of the underlying manifold. The algebraic requirement $\text{proj}_X f(x) = x$ for all $x$ ensures that the tangent vector chosen correctly corresponds to the point at which it is attached.

[2] If a smooth choice of basis vectors existed, it would define a smooth non-vanishing vector field. On the sphere, by the *hairy ball theorem*, all smooth vector fields vanish in at least one point, so no such bases exist.

**Proposition 2.** *Let $X$ be a manifold and* $\mathrm{emb} : X \to \mathbb{R}^p$ *be a smooth embedding. Let $f$ be a Gaussian vector field (as defined in Definition 1), and let $f_{\mathrm{emb}} : \mathrm{emb}(X) \to \mathbb{R}^p$ be its pushforward along the embedding. Then $f_{\mathrm{emb}}$ is a vector-valued Gaussian process in the Euclidean sense.*

All proofs in this work can be found in Appendix A. Having established the notion of a vector-valued Gaussian process on a smooth manifold, we proceed to deduce what mathematical objects play the role of a mean function and kernel, so that it is clear what ingredients are needed to construct and determine such a process.

The former is clear: the mean of a Gaussian vector field should be an ordinary vector field, and will determine the mean vector at all finite-dimensional marginals. The kernel, on the other hand, is less obvious: because distinct tangent vectors live in different tanget spaces, it is unclear whether or not a Gaussian vector field is characterized by an appropriate notion of a matrix-valued kernel, or by something else. Now, we define the right notion for kernel in this setting.

**Definition 3.** *We call a symmetric function $k : T^*X \times T^*X \to \mathbb{R}$ fiberwise bilinear if for all pairs of points $x, x' \in X$*

$$k(\lambda\alpha_x + \mu\beta_x, \gamma_{x'}) = \lambda k(\alpha_x, \gamma_{x'}) + \mu k(\beta_x, \gamma_{x'}) \tag{3}$$

*holds for any $\alpha_x, \beta_x \in T_x^*X$, $\gamma_{x'} \in T_{x'}^*X$ and $\lambda, \mu \in \mathbb{R}$, and* positive semi-definite *if for any set of covectors $\alpha_{x_1}, \ldots, \alpha_{x_n} \in T^*X$, we have $\sum_{i=1}^n \sum_{j=1}^n k(\alpha_{x_i}, \alpha_{x_j}) \geq 0$. We call a symmetric fiberwise bilinear positive semi-definite function a* cross-covariance kernel.

This coordinate-free function should be viewed as analogous to $((x, \boldsymbol{v}), (x', \boldsymbol{v}')) \mapsto \boldsymbol{v}^T \mathbf{K}_{x,x'} \boldsymbol{v}'$ in the Euclidean setting, where $\boldsymbol{v}, \boldsymbol{v}'$ multiply the matrix-valued kernel from both sides. Its coordinate representation, which more closely matches the Euclidean case, will be explored in the sequel. To show that this is indeed the right notion, we prove the following result.

**Theorem 4.** *The system of marginal distributions of a Gaussian vector field on a smooth manifold $X$ is uniquely determined by a mean vector field $\mu : X \to TX$ and a cross-covariance kernel $k : T^*X \times T^*X \to \mathbb{R}$. Moreover, this correspondence is one-to-one.*

By virtue of defining and characterizing all Gaussian vector fields, Theorem 4 assures us the definition of a kernel introduced is the correct mathematical notion. The constructions presented here are all intrinsic or, in other words, coordinate-free, and do not involve the use of bases. To understand how to perform numerical calculations with these kernels we proceed to study their coordinate representations with respect to a specific choice of basis.

### 2.3 Matrix-valued Kernels

In Section 2.2, we defined what a Gaussian vector field on a manifold is. However, by nature of the manifold setting, the resulting objects are more abstract than usual and do not describe how it can be represented numerically. We now develop a point of view suitable for this task.

To this end, we introduce a *frame $F$* on $X$, also known as a *gauge* in physical literature, which is a collection of (not necessarily smooth) vector fields $e_1, \ldots, e_d$ on $X$ such that at each point $x \in X$, the set of vectors $e_1(x), \ldots, e_d(x)$ forms a basis of $T_xX$. The frame allows us to express a vector field $f$ on $X$ as simply a vector-valued function $\boldsymbol{f} = (f^1, \ldots, f^d) : X \to \mathbb{R}^d$, such that $f(x) = \sum_{i=1}^d f^i(x) e_i(x)$ for all $x \in X$. The corresponding *coframe $F^*$* is defined as a collection $e^1, \ldots, e^d$ of covector fields (one-forms) on $X$ such that $\langle e^i(x)|e_j(x)\rangle = \delta_{ij}$ for all $x \in X$, where $\delta_{ij}$ is the Kronecker delta. In the following proposition, we show that if $f$ is a Gaussian vector field on $X$ (in the sense of Definition 1), then the corresponding vector representation $\boldsymbol{f}$ expressed in a given frame is a vector-valued GP in the standard sense.

**Proposition 5.** *Let $f$ be a Gaussian vector field defined on $X$ with cross-covariance kernel $k : T^*X \times T^*X \to \mathbb{R}$. Given a frame $F = (e_1, \ldots, e_d)$ on $X$, define $\boldsymbol{f} : X \to \mathbb{R}^d$ as above. Then $\boldsymbol{f}$ is a vector-valued GP in the usual sense with kernel $\mathbf{K}_F : X \times X \to \mathbb{R}^{d \times d}$ given by*

$$\mathbf{K}_F(x, x') = \begin{bmatrix} k(e^1(x), e^1(x')) & \ldots & k(e^1(x), e^d(x')) \\ \vdots & \ddots & \vdots \\ k(e^d(x), e^1(x')) & \ldots & k(e^d(x), e^d(x')) \end{bmatrix}, \tag{4}$$

*where $(e^i)$, with raised indices, is the coframe corresponding to $(e_i)$. Conversely, given a vector-valued GP $\boldsymbol{f} = (f^1, \ldots, f^d) : X \to \mathbb{R}^d$ and a frame $F = (e_1, \ldots, e_d)$ on $X$, $f(\cdot) := \sum_{i=1}^d f^i(\cdot)e_i(\cdot)$ defines a Gaussian vector field on $X$.*

This result shows precisely how numerical representations of a Gaussian vector field depends on the choice of frame. While this representation is not *invariant* under this choice, it is *equivariant*, meaning that a transformation in the frame results in an appropriate transformation of the kernel. To make this notion precise, we introduce a matrix subgroup $G = \mathrm{GL}(d, \mathbb{R})$, called the *gauge group*, that acts on $\mathbb{R}^d$ by a standard matrix-vector multiplication. Given two frames $F, F'$ on $X$, an abstract vector $f_x \in T_x X$ has two vector representations $\boldsymbol{f}_x, \boldsymbol{f}_x'$ in the respective frames. We say that $F'$ is obtained from $F$ by a *gauge transformation* with respect to a matrix field $\mathbf{A} : X \to G \subseteq \mathbb{R}^{d \times d}$, if

$$\boldsymbol{f}_x' = \mathbf{A}(x)\boldsymbol{f}_x \tag{5}$$

holds for all $x \in X$, and we write $F' = \mathbf{A}F$. Note that $\mathbf{A}(x)$ need not be smooth in $x$. We see that the gauge transformation is therefore just a linear change of basis of the frame $F$ at each point for which one can identify vectors in $T_x X$ as elements in $\mathbb{R}^d$. The corresponding gauge dependant matrix-valued kernels must also respect this transformation rule, a statement of *gauge independence*.

**Corollary 6.** *Let $F$ be a frame on $X$ and $\mathbf{K}_F : X \times X \to \mathbb{R}^{d \times d}$ be the corresponding matrix representation* (4) *of a cross-covariance kernel $k : T^*X \times T^*X \to \mathbb{R}$. This satisfies the* equivariance condition

$$\mathbf{K}_{\mathbf{A}F}(x, x') = \mathbf{A}(x)\mathbf{K}_F(x, x')\mathbf{A}(x')^T, \tag{6}$$

*where $\mathbf{A} : X \to G \subseteq \mathbb{R}^{d \times d}$ is a gauge transformation. All cross-covariance kernels in the sense of Proposition 4 arise this way.*

Hence, one way to define a Gaussian vector field on a manifold is to find a gauge independent kernel. In summary, we have described Gaussian vector fields in a coordinate-free differential-geometric language, and deduced enough properties to confirm the objects defined truly deserve to be called GPs. In doing so, we have both introduced the necessary formalism to the GP community, and obtained a recipe for defining kernels, through a simple condition atop standard matrix-valued kernels. To proceed towards practical machine learning methods, we therefore study techniques for constructing such kernels explicitly.

## 3 Model Construction and Bayesian Learning for Riemannian Manifolds

In Section 2, we introduced a notion of a Gaussian vector field. We now study how to use vector fields for machine learning purposes. This entails two primary issues: (a) how to construct practically useful kernels, and (b) once a kernel is constructed, how to train Gaussian processes.

To construct a Gaussian vector field prior, the preceding theory tells us that we need to specify a mean vector field and a cross-covariance kernel. From the definition, it is not at all obvious how to specify a natural kernel, and experience with the scalar-valued case—where the innocuous-looking geodesic squared exponential kernel is generally not positive semi-definite on most Riemannian manifolds [13]—suggests that the problem is delicate, i.e., simply guessing the kernel's form is unlikely to succeed. Our goal, therefore, is to introduce a general construction for building wide classes of kernels from simple building blocks.

The same issues are present if we consider variational approximations to posterior GPs, such as the inducing point framework of Titsias [43]: these are formulated using matrix-vector expressions involving kernel matrices, and it is important for the approximate posterior covariance to be gauge independent in order to lead to a valid approximate process. We proceed to address these issues.

### 3.1 Projected Kernels

Here, we introduce a general technique for defining cross-covariance kernels $k : T^*X \times T^*X \to \mathbb{R}$ and for working with such functions numerically. Section 2 gives us a promising strategy to construct a suitable kernel—namely, it suffices to find a *gauge independent matrix-valued kernel*. At first glance, it is not obvious how to construct such a kernel in the manifold setting. On many manifolds, such as the sphere, owing to the hairy ball theorem, every frame must be discontinuous: therefore, constructing a continuous kernel in such a choice of frame appears difficult.

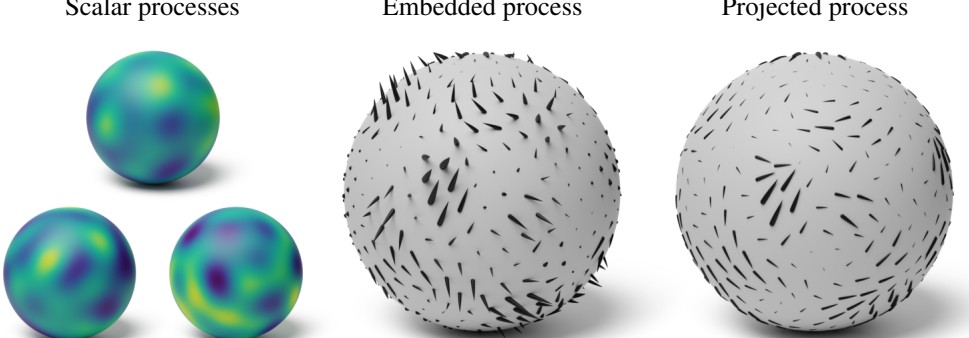

Scalar processes        Embedded process        Projected process

Figure 2: Illustration of the construction process of the projected process. The manifold $\mathbb{S}^2$ is embedded into $\mathbb{R}^3$. Three identical scalar GPs (left) are placed on the manifold. These three scalar GPs are combined to construct a vector-valued GP in the ambient Euclidean space (center). This GP is then projected onto the tangent space of $\mathbb{S}^2$ as a subspace of the tangent space of $\mathbb{R}^3$ (right).

To both get around these obstacles, and aid numerical implementation, we propose to isometrically embed the manifold into Euclidean space.[3] Doing so greatly simplifies these issues by virtue of making it possible to represent the manifold using a single global coordinate system. On the other hand, the main trade-off from this choice is that by its extrinsic nature, the construction can make theoretical analysis more difficult. To proceed, we need two ingredients.

1. An isometric embedding $\mathrm{emb} : X \to \mathbb{R}^{d'}$ of the manifold.
2. A vector-valued Gaussian process $\boldsymbol{f}' : X \to \mathbb{R}^{d'}$ in the standard sense.[4]

A simple choice which reflects the geometry of the manifold is to take $\boldsymbol{f}'$ to be $d'$ independent scalar-valued GPs on $X$.

By standard results in differential geometry, any smooth map $\phi : X \to X'$ between two manifolds induces a corresponding linear map on the tangent spaces $\mathrm{d}_x\phi : T_xX \to T_{\phi(x)}X'$, which can loosely be thought of as mapping $\phi$ to its first-order Taylor expansion at $x$. Thus, an embedding $\mathrm{emb} : X \to \mathbb{R}^{d'}$, induces a map $\mathrm{d}_x\mathrm{emb} : T_xX \to T_{\mathrm{emb}(x)}\mathbb{R}^{d'}$. Now fixing a frame $F$ on $X$, each tangent space $T_xX$ can be identified with $\mathbb{R}^d$, so without loss of generality, the map $\mathrm{d}_x\mathrm{emb}$ can be expressed simply as a position-dependent matrix $\mathbf{P}_x^T \in \mathbb{R}^{d' \times d}$. Taking the transpose, we obtain $\mathbf{P}_x \in \mathbb{R}^{d \times d'}$, which we call the *projection matrix*. The desired Gaussian vector field on $X$, with respect to $F$, is then constructed as $\boldsymbol{f}(x) = \mathbf{P}_x\boldsymbol{f}'(x)$. This procedure is illustrated in Figure 2: there we see that to get a vector field on an $\mathbb{R}^3$-embedded sphere, we may take a vector-valued function on it and project its values to make vectors tangential to this sphere, thus obtaining a valid vector field. Since the projection operator preserves smoothness and since we can take a smooth vector-valued GP to begin with, it is clear that this approach may be used to build smooth vector fields.

We prove that (a) the resulting expression is, indeed, a kernel, and that (b) no expressivity is lost via the construction because all cross-covariance kernels arise this way.

**Proposition 7.** *Let $(X, g)$ be a Riemannian manifold,* $\mathrm{emb} : X \to \mathbb{R}^{d'}$ *be an isometric embedding and $F$ be a frame on $X$. We denote by $\mathbf{P}_{(\cdot)} : X \to \mathbb{R}^{d \times d'}$ the associated projection matrix under $F$, and let $\boldsymbol{f}' : X \to \mathbb{R}^{d'}$ be any vector-valued Gaussian process with matrix-valued kernel $\boldsymbol{\kappa} : X \times X \to \mathbb{R}^{d' \times d'}$. Then, the vector-valued function $\boldsymbol{f} = \mathbf{P}\boldsymbol{f}'$ defines a Gaussian vector field $\boldsymbol{f}$ on $X$ using the construction in Proposition 5, whose kernel under the frame $F$ has matrix representation*

$$\mathbf{K}_F(x, x') = \mathbf{P}_x \boldsymbol{\kappa}(x, x') \mathbf{P}_{x'}^T. \tag{7}$$

---

[3]An embedding $\mathrm{emb} : X \to \mathbb{R}^{d'}$ is called *isometric* if it preserves the metric tensor. By Nash's Theorem [26], such embeddings exist for any $d$-dimensional manifold, with an embedded dimension $d' \leq 2d + 1$.

[4]We emphasize again that $\boldsymbol{f}'$ is *not* a Gaussian vector field because it is not a random section. In particular, note that $d' > d$ for most embeddings.

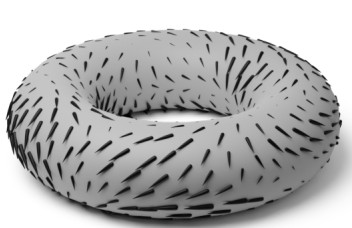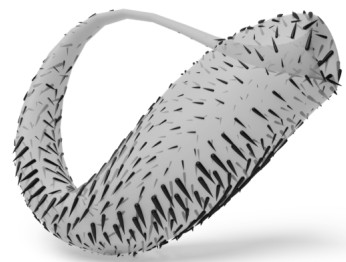

Figure 3: Random samples from Gaussian processes with gauge independent projected kernels, on the torus and Klein bottle, respectively. The latter is a non-orientable manifold: the ability to handle such cases highlights the generality of projected kernels.

*Moreover, all cross-covariance kernels $k : T^*X \times T^*X \to \mathbb{R}$ arise this way. We call a kernel defined this way a* projected kernel.

To construct these kernels we require scalar-valued kernels on manifolds to use as a basic building block. These are studied in the general Riemannian setting by Lindgren et al. [28] and Borovitskiy et al. [2]: relying on these kernels is the only reason we require the Riemannian structure. It is also possible to obtain such kernels using embeddings, following Lin et al. [27]. Similar techniques to those we consider are used by Freeden and Schreiner [14] to construct vector-valued zonal kernels on the sphere: in contrast, we work with arbitrary manifolds. The projection kernel idea is a very general way to build kernels for vector fields by combining scalar kernels, but effective scalar kernels, naturally, rely on Riemannian structure.[5] Figure 3 shows random samples from Gaussian processes constructed with the described kernels.

The projected kernel construction both makes it easy to define cross-covariance kernels on general manifolds, and describes a straightforward way to implement them numerically by representing the embedded manifold in coordinates and calculating the resulting matrix-vector expressions. The constructed kernel depends on the embedding, but can be transformed appropriately if switching to a different embedding. Embeddings, in turn, are available for most manifolds of practical interest, and are obtained automatically for manifolds approximated numerically as meshes. Everything described is constructive and fully compatible with the modern automatic-differentiation-based machine learning toolkit, and most operations for constructing and/or sampling from specialized priors [44, 22, 23], including on spaces such as the sphere where specific analytic tools are available [6, 8, 10, 9]. With these kernels in hand, we thus proceed to study training methods.

### 3.2   Gauge Independent Variational Approximations

We now discuss variational inference for training GPs in the Riemannian vector field setting. Approximations, such as the inducing-point framework by Titsias [43] and Hensman et al. [17], approximate the posterior GP with another GP, termed the *variational approximation*. The latter is typically constructed by specifying a multivariate Gaussian at a set of test locations with a parameterized mean and kernel matrix. For example, Opper and Archambeau [31] consider $N(\mathbf{m}, \mathbf{S})$, where

$$\mathbf{m} = \mathbf{K}_{(\cdot)\boldsymbol{z}}(\mathbf{K}_{\boldsymbol{z}\boldsymbol{z}} + \boldsymbol{\Sigma})^{-1}\boldsymbol{\mu} \qquad \mathbf{S} = \mathbf{K}_{(\cdot,\cdot)} - \mathbf{K}_{(\cdot)\boldsymbol{z}}(\mathbf{K}_{\boldsymbol{z}\boldsymbol{z}} + \boldsymbol{\Sigma})^{-1}\mathbf{K}_{\boldsymbol{z}\boldsymbol{z}}(\mathbf{K}_{\boldsymbol{z}\boldsymbol{z}} + \boldsymbol{\Sigma})^{-1}\mathbf{K}_{\boldsymbol{z}(\cdot)}. \quad (8)$$

The variational parameters include a set of inducing locations $\boldsymbol{z}$, a mean vector $\boldsymbol{\mu}$, and a block-diagonal cross-covariance matrix $\boldsymbol{\Sigma}$. Training proceeds by optimizing these parameters to minimize the Kullback–Leibler divergence of the variational distribution from the true posterior, typically using mini-batch stochastic gradient descent.

In the last decade, a wide and diverse range of inducing point approximations suited for many different settings have been proposed [43, 31, 25, 44, 49]. The vast majority of them employ coordinate-

---

[5]The structure of a smooth manifold is not rigid enough to define natural kernels. For instance, the smooth structure of the sphere is indistinguishable from the smooth structure of an ellipsoid or even of the dragon manifold from Borovitskiy et al. [2], but their Riemannian structures differ considerably.

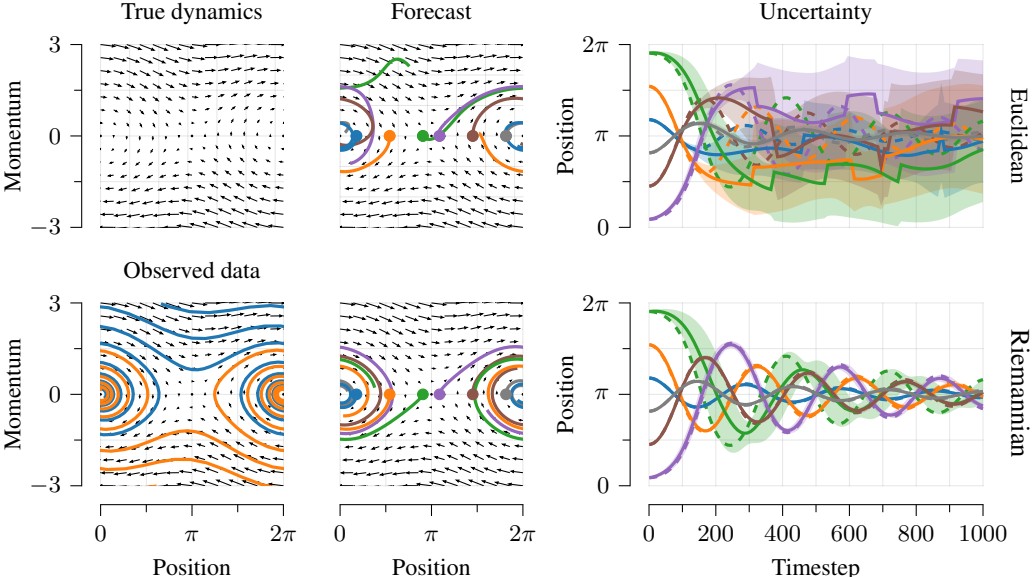

Figure 4: Upper left: pendulum with friction state space. Lower left: Two rollouts used to train the GP. Upper middle: State space plot of rollouts from a standard Euclidean GP condtioned on the training data. Upper right: temporal plot of rollouts from a standard Euclidean vector GP. Solid line is the true rollout, dashed line and shade is the mean and $\pm$ 1 std of the GP rollouts. Lower middle and right: same for a geometric manifold vector field kernel on $\mathbb{S}^1 \times \mathbb{R}$.

dependent matrix-vector expressions. This raises the question, which of these constructions can be adapted to define valid variational approximations in the vector field setting?

To proceed, one can choose a frame and formulate a given variational approximation using matrices defined with respect to this frame. To ensure well-definedness, one must ensure that all these matrices, such as the kernel matrix and the variational parameter $\Sigma$ in (8), are gauge independent. These considerations can be simplified adopting the pathwise view of GPs, and examining the random variables directly. For example, the variational approximation of Opper and Archambeau [31] shown previously in (8) can be reinterpreted pathwise as the GP

$$(f \mid y)(\cdot) \approx f(\cdot) + \mathbf{K}_{(\cdot)\boldsymbol{z}}(\mathbf{K}_{\boldsymbol{zz}} + \boldsymbol{\Sigma})^{-1}(\boldsymbol{\mu} - f(\boldsymbol{z}) - \boldsymbol{\varepsilon}) \qquad \boldsymbol{\varepsilon} \sim \mathrm{N}(\mathbf{0}, \boldsymbol{\Sigma}) \qquad (9)$$

where we view the matrices $\mathbf{K}_{(\cdot)\boldsymbol{z}}, \mathbf{K}_{\boldsymbol{zz}}, \boldsymbol{\Sigma}$ as *linear operators* between direct sums of tangent spaces: $\mathbf{K}_{(\cdot)\boldsymbol{z}} : T_{z_1} X \oplus \ldots \oplus T_{z_m} X \to T_{(\cdot)} X$ and $\mathbf{K}_{\boldsymbol{zz}}, \boldsymbol{\Sigma} : T_{z_1} X \oplus \ldots \oplus T_{z_m} X \to T_{z_1} X \oplus \ldots \oplus T_{z_m} X$. By virtue of being defined at the level of vector fields using components that are all intrinsically valid, the posterior covariance of the resulting variational approximation is *automatically* gauge independent. Hence, checking gauge independence is then equivalent to deducing the domains and ranges of these operators from their coordinate representations, and checking if they are compatible. This applies to any variational family that can be constructed in the given manner.

The vast majority of inducing point constructions can be interpreted in this manner and thus extend readily to the Riemannian vector field setting by simply representing the necessary matrices in a chosen frame. In particular, the classical approach of Titsias [43] is gauge independent.

# 4 Illustrated Examples

Here, we showcase a number of examples that illustrate potential use cases of the models developed.

## 4.1 Dynamical Systems Modeling

Here, we show how Gaussian vector fields can be used to learn the equations of motion of a physical system—an important task in imitation learning, model-based reinforcement learning, and robotics. GPs are an attractive model class in this area owing to their ability to represent and propagate

uncertainty, which enables them to separate what is known about an environment from what is not, thereby driving data-efficient exploration.

For a prototype physical system, we consider an ideal pendulum, whose configuration space is the circle $\mathbb{S}^1$, representing the angle of the pendulum, with zero being at the bottom of the loop, and whose position-momentum state-space is the cylinder $\mathbb{S}^1 \times \mathbb{R}$. We consider conservative dynamics with additional friction applied at the pivot. Since this system is non-conservative, we cannot just learn the Hamiltonian of the system, but must learn the vector field over the state space that defines the dynamics of the system. The true dynamics of the system are given by the differential equations

$$\mathcal{H} = \frac{p^2}{2ml^2} + mgl(1 - \cos(q)) \qquad \frac{\mathrm{d}q}{\mathrm{d}t} = \frac{\partial \mathcal{H}}{\partial p} \qquad \frac{\mathrm{d}p}{\mathrm{d}t} = -\frac{\partial \mathcal{H}}{\partial q} - \frac{b}{m}p, \qquad (10)$$

where $\mathcal{H}$ is the Hamiltonian of the system defining the conservative part of the dynamics, $q$ and $p$ are the position and momentum of the pendulum, $m$ is the mass, $l$ is the length, $g$ is the gravitational field strength and $b$ is a friction parameter. Experimental details can be found in Appendix B.

To learn this model, we initialise the system at two start points, and evolve the system using leapfrog integration. From these observations of position, we backward Euler integrate the momentum of the system, and from these position-momentum trajectories we estimate observations of the dynamics field. Using these observations, we condition a sparse GP. The result is an estimate of the system dynamics with suitable uncertainty estimates. In order to compute rollouts of these dynamics, we use pathwise sampling of this sparse GP [47, 48] for speed together with leapfrog integration.

Results can be seen in Figure 4. While the Euclidean GP performs reasonably well at the start of the rollouts, once the trajectory crosses the discontinuity caused by looping the angle back around to zero, the system starts to make incoherent predictions: this is due to the discontinuity arising from wrap-around condition of the angle. The manifold vector-valued GP does not have this issue as the learned and sampled dynamics fields are continuous throughout the state-space.

## 4.2 Weather Modeling

In this experiment, we show how vector-valued GPs on manifolds can be used in the context of meteorology, where geometric information often plays an important role in accurately modeling global weather fields [46, 38, 19]. *Data assimilation* in numerical weather forecasting refers to the practice of using observed data to update predictions of the atmosphere closer to the truth. Uncertainty plays a critical role here: it is not usually possible to observe the weather at all locations on the globe simultaneously, and taking into account observation uncertainty is crucial in numerical weather forecasting during the data assimilation step [24, 36]. In this section, we explore Gaussian processes as a tool for carrying out global interpolation of wind fields, while simultaneously performing uncertainty quantification, mirroring *optimal interpolation* techniques in data assimilation [20].

We consider a simplified setting, where the goal is to interpolate the wind velocity observed by the Aeolus satellite [37], which uses LiDAR sensors to measure wind velocity directly. To mimic this setting, we use an hour of the Aeolus satellite track during the period 2019/01/01 09:00-10:00 for the input locations and the wind velocity data (10m above ground) from the ERA5 atmospheric reanalysis data [18] interpolated at these locations, to simulate measurements taken from the Aeolus satellite. We subtract the weekly historical average wind velocity from the observations, before training the GP models, where the historical mean is computed from the hourly wind data (10 m above ground) from the WeatherBench dataset [35], available from 1979–2018. Further details can be found in Appendix B. We compare the results of a Matérn-3/2 manifold vector-valued GP regression model fitted on the wind anomaly observations along the Aeolus trajectory, with the results from a Euclidean Matérn-3/2 multi-output GP trained on the same data, except projected onto a latitude-longitude map.

Results are shown in Figure 5, where the benefits of using a manifold vector-valued GP become clear. When the satellite crosses the left/right boundary in the lat/lon projection, the outputs from the Euclidean vector-valued GP give rise to a spurious discontinuity in the uncertainty along the solid pink line. In addition, predictions become less certain in the Euclidean case as the satellite approaches the poles, which is simply an artifact of the distortion caused by projecting the spherical data onto the plane. By construction, the manifold vector-valued GP is able to avoid both of these issues, resulting in a more realistic prediction with much more uniform uncertainty along the satellite trajectory from pole to pole. In addition, the predictions from the manifold GP are more certain overall, due to the useful structural bias embedded in the kernel.

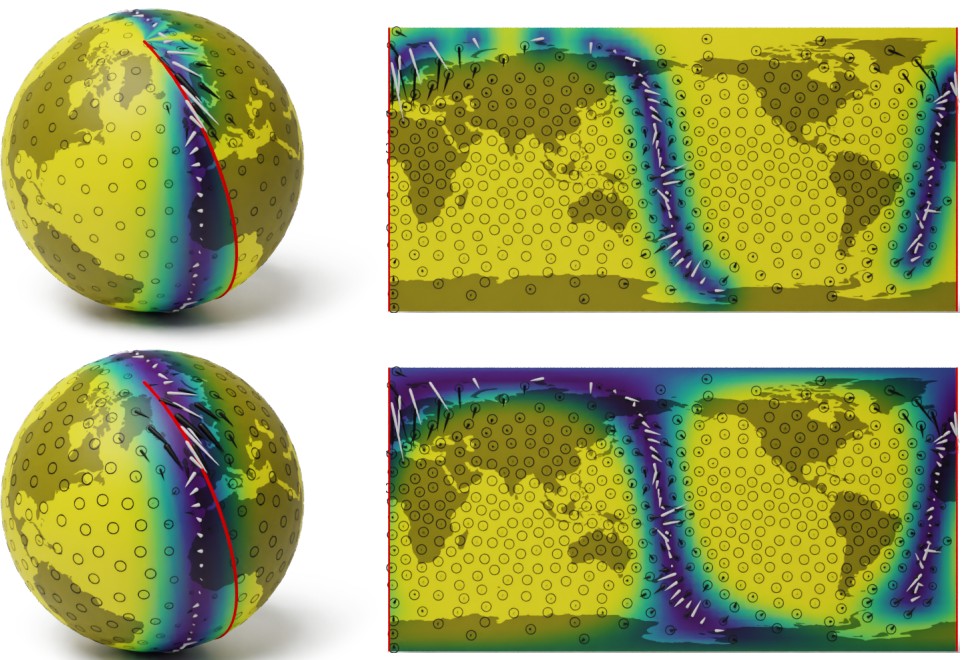

Figure 5: Top row: Euclidean GP trained on wind measurements along the chosen Aeolus satellite trajectory, viewed as deviation from normal with respect to the historical average vector field. White arrows are the satellite measurements, black arrows and ellipsoids are the posterior mean and cross-covariance of the vector field, colors indicate the posterior standard deviation norm, and the solid red line indicates the latitudinal boundary when the sphere is projected onto the plane using the lat/lon projection. Bottom row: Same as above except using a manifold kernel on $\mathbb{S}^2$.

## 5 Conclusion

In this paper, we propose techniques that generalize Gaussian processes to model vector fields on Riemannian manifolds. This is done by first providing a well-defined notion of such processes on manifolds and then introducing an explicit method to construct them in a way that respects the underlying geometry. By virtue of satisfying the key condition of gauge independence, our construction is coordinate-free and thus meaningful on manifolds. In addition to this, we extend standard Gaussian process training methods, such as variational inference, to this setting, and verify that such methods are also compatible with gauge independence. This theoretical work gives practitioners additional tools for stochastic modeling of vector fields on manifolds. As such, its societal impact will be mainly determined by the applications that belong to the domain of future work. We demonstrate our techniques on a series of examples in modeling dynamical systems and weather science, and show that incorporating geometric structural bias into probabilistic modeling is beneficial in these settings to obtain coherent predictions and uncertainties.

## 6 Acknowledgements and Funding Disclosure

Michael is supported by the EPSRC Centre for Doctoral Training in Modern Statistics and Statistical Machine Learning (EP/S023151/1). Section 3 and Appendix A of this paper were solely financially supported by the RSF grant №21-11-00047 (https://rscf.ru/en/project/21-11-00047/), but was contributed to by all authors in equal proportion with the remaining work. Other sections were supported by the remaining funding. We also acknowledge support from Huawei Research. We thank Peter Dueben for useful suggestions in Section 4.2, including, in particular, making us aware of the Aeolus satellite. We are grateful to Nick Sharp for introducing us to the process used for making the three-dimensional figures appearing in this work.

