# References

[1] V. Borovitskiy, I. Azangulov, A. Terenin, P. Mostowsky, M. P. Deisenroth, and N. Durrande. Matérn Gaussian Processes on Graphs. In *Artificial Intelligence and Statistics*, 2021. Cited on page 1.

[2] V. Borovitskiy, A. Terenin, P. Mostowsky, and M. P. Deisenroth. Matern Gaussian Processes on Riemannian Manifolds. In *Advances in Neural Information Processing Systems*, 2020. Cited on pages 1, 2, 7, 22–24.

[3] E. Brochu, V. M. Cora, and N. de Freitas. A Tutorial on Bayesian Optimization of Expensive Cost Functions, with Application to Active User Modeling and Hierarchical Reinforcement Learning. Technical report, University of British Columbia, 2009. Cited on page 1.

[4] Y. Canzani. Analysis on Manifolds via the Laplacian. Lectures notes, Harvard University, 2013. Cited on page 22.

[5] T. Cohen, M. Weiler, B. Kicanaoglu, and M. Welling. Gauge Equivariant Convolutional Networks and the Icosahedral CNN. In *International Conference on Machine Learning*, 2019. Cited on page 2.

[6] P. E. Creasey and A. Lang. Fast Generation of Isotropic Gaussian Random Fields on the Sphere. *Monte Carlo Methods and Applications*, 24(1):1–11, 2018. Cited on page 7.

[7] M. P. Deisenroth and C. E. Rasmussen. PILCO: A Model-Based and Data-Efficient Approach to Policy Search. In *International Conference on Machine Learning*, 2011. Cited on page 1.

[8] V. Dutordoir, N. Durrande, and J. Hensman. Sparse Gaussian Processes with Spherical Harmonic Features. In *International Conference on Machine Learning*, 2020. Cited on page 7.

[9] X. Emery, R. Furrer, and E. Porcu. A Turning Bands Method for Simulating Isotropic Gaussian Random Fields on the Sphere. *Statistics and Probability Letters*, 144:9–15, 2019. Cited on page 7.

[10] X. Emery and E. Porcu. Simulating Isotropic Vector-valued Gaussian Random Fields on the Sphere through Finite Harmonics Approximations. *Stochastic Environmental Research and Risk Assessment*, 33(2):1659–1667, 2019. Cited on page 7.

[11] M. Émery. *Stochastic Calculus in Manifolds*. Springer, 2012. Cited on page 2.

[12] European Centre for Medium-Range Weather Forecasts. The ESA ADM-Aeolus Doppler Wind Lidar Mission. 2017. URL: https://www.ecmwf.int/sites/default/files/elibrary/2016/1685 1-esa-adm-aeolus-doppler-wind-lidar-mission-status-and-validation-strategy.pdf. Cited on page 24.

[13] A. Feragen, F. Lauze, and S. Hauberg. Geodesic Exponential Kernels: When Curvature and Linearity Conflict. In *Conference on Computer Vision and Pattern Recognition*, 2015. Cited on pages 2, 5.

[14] W. Freeden and M. Schreiner. *Spherical Functions of Mathematical Geosciences: A Scalar, Vectorial, and Tensorial Setup*. Springer, 2008. Cited on page 7.

[15] P. D. Haan, M. Weiler, T. Cohen, and M. Welling. Gauge Equivariant Mesh CNNs: Anisotropic Convolutions on Geometric Graphs. In *International Conference on Learning Representations*, 2021. Cited on page 2.

[16] E. Hairer, C. Lubich, and G. Wanner. *Geometric Numerical Integration: Structure-preserving Algorithms for Ordinary Differential Equations*. Springer, 2006. Cited on page 23.

[17] J. Hensman, N. Fusi, and N. D. Lawrence. Gaussian Processes for Big Data. In *Uncertainty in Artificial Intelligence*, 2013. Cited on page 7.

[18] H. Hersbach. The ERA5 Atmospheric Reanalysis. In *American Geophysical Union Fall Meeting Abstracts*, 2016. Cited on page 9.

[19] I. L. Jover. *Geometric Deep Learning for Medium Range Weather Prediction*. Master's thesis, École Polytechnique Fédérale de Lausanne, 2020. Cited on page 9.

[20] E. Kalnay. *Atmospheric Modeling, Data Assimilation and Predictability*. Cambridge University Press, 2003. Cited on page 9.

[21] A. Krause, A. Singh, and C. Guestrin. Near-optimal Sensor Placements in Gaussian Processes: Theory, Efficient Algorithms and Empirical Studies. *Journal of Machine Learning Research*, 9(8):235–284, 2008. Cited on page 1.

[22] M. Lange-Hegermann. Algorithmic Linearly Constrained Gaussian Processes. *Advances in Neural Information Processing Systems*, 2018. Cited on page 7.

[23]  M. Lange-Hegermann. Linearly Constrained Gaussian Processes with Boundary Conditions. In *Artificial Intelligence and Statistics*, 2021. Cited on page 7.

[24]  K. Law, A. Stuart, and K. Zygalakis. *Data Assimilation*. Springer, 2015. Cited on page 9.

[25]  M. Lázaro-Gredilla and A. R. Figueiras-Vidal. Inter-domain Gaussian Processes for Sparse Inference Using Inducing Features. In *Advances in Neural Information Processing Systems*, 2009. Cited on page 7.

[26]  J. M. Lee. Smooth Manifolds. In *Introduction to Smooth Manifolds*. Springer, 2013. Cited on pages 1, 6.

[27]  L. Lin, N. Mu, P. Cheung, and D. Dunson. Extrinsic Gaussian Processes for Regression and Classification on Manifolds. *Bayesian Analysis*, 14(3):887–906, 2019. Cited on page 7.

[28]  F. Lindgren, H. Rue, and J. Lindström. An Explicit Link between Gaussian Fields and Gaussian Markov Random Fields: the Stochastic Partial Differential Equation Approach. *Journal of the Royal Statistical Society: Series B (Statistical Methodology)*, 73(4):423–498, 2011. Cited on page 7.

[29]  A. Mallasto and A. Feragen. Wrapped Gaussian Process Regression on Riemannian Manifolds. In *Conference on Computer Vision and Pattern Recognition*, 2018. Cited on page 2.

[30]  A. Mallasto, S. Hauberg, and A. Feragen. Probabilistic Riemannian Submanifold Learning with Wrapped Gaussian Process Latent Variable Models. In *Artificial Intelligence and Statistics*, 2019. Cited on page 2.

[31]  M. Opper and C. Archambeau. The Variational Gaussian Approximation Revisited. *Neural Computation*, 21(3):786–792, 2009. Cited on pages 7, 8.

[32]  M. A. Osborne, R. Garnett, and S. J. Roberts. Gaussian Processes for Global Optimization. In *International Conference on Learning and Intelligent Optimization*, 2009. Cited on page 1.

[33]  A. Rahimi and B. Recht. Random Features for Large-scale Kernel Machines. In *Advances in Neural Information Processing Systems*, 2008. Cited on page 22.

[34]  C. E. Rasmussen and M. Kuss. Gaussian Processes in Reinforcement Learning. In *Advances in Neural Information Processing Systems*. 2004. Cited on page 1.

[35]  S. Rasp, P. D. Dueben, S. Scher, J. A. Weyn, S. Mouatadid, and N. Thuerey. WeatherBench: A Benchmark Data Set for Data-driven Weather Forecasting. *Journal of Advances in Modeling Earth Systems*, 12(11):e2020MS002203, 2020. Cited on pages 9, 23.

[36]  S. Reich and C. Cotter. *Probabilistic Forecasting and Bayesian Data Assimilation*. Cambridge University Press, 2015. Cited on page 9.

[37]  O. Reitebuch. The Spaceborne Wind LiDAR Mission ADM-Aeolus. In *Atmospheric Physics*, pages 815–827. 2012. Cited on page 9.

[38]  S. Scher and G. Messori. Spherical Convolution and Other Forms of Informed Machine Learning for Deep Neural Network Based Weather Forecasts. *arXiv:2008.13524*, 2020. Cited on page 9.

[39]  B. Shahriari, K. Swersky, Z. Wang, R. P. Adams, and N. De Freitas. Taking the Human out of the Loop: A Review of Bayesian Optimization. *Proceedings of the IEEE*, 104(1):148–175, 2016. Cited on page 1.

[40]  D. W. Stroock. *An Introduction to the Analysis of Paths on a Riemannian Manifold*. American Mathematical Society, 2000. Cited on page 2.

[41]  D. J. Sutherland and J. Schneider. On the Error of Random Fourier Features. In *Uncertainty in Artificial Intelligence*, 2015. Cited on page 22.

[42]  T. Tao. *An Introduction to Measure Theory*. American Mathematical Society, 2011. Cited on page 16.

[43]  M. K. Titsias. Variational Learning of Inducing Variables in Sparse Gaussian Processes. In *Artificial Intelligence and Statistics*, 2009. Cited on pages 5, 7, 8.

[44]  M. van der Wilk, V. Dutordoir, S. John, A. Artemev, V. Adam, and J. Hensman. A Framework for Interdomain and Multioutput Gaussian Processes. *arXiv:2003.01115*, 2020. Cited on page 7.

[45]  M. Weiler, P. Forré, E. Verlinde, and M. Welling. Coordinate Independent Convolutional Networks—Isometry and Gauge Equivariant Convolutions on Riemannian Manifolds. *arXiv:2106.06020*, 2021. Cited on page 2.

[46] J. A. Weyn, D. R. Durran, and R. Caruana. Improving Data-Driven Global Weather Prediction Using Deep Convolutional Neural Networks on a Cubed Sphere. *Journal of Advances in Modeling Earth Systems*, 12(9):e2020MS002109, 2020. Cited on page 9.

[47] J. T. Wilson, V. Borovitskiy, A. Terenin, P. Mostowsky, and M. P. Deisenroth. Pathwise Conditioning of Gaussian Processes. In *International Conference on Machine Learning*, 2020. Cited on pages 2, 9, 22, 23.

[48] J. T. Wilson, V. Borovitskiy, A. Terenin, P. Mostowsky, and M. P. Deisenroth. Pathwise Conditioning of Gaussian Processes. *Journal of Machine Learning Research*, 22(105):1–47, 2021. Cited on pages 2, 9, 22, 23.

[49] L. Wu, A. Miller, L. Anderson, G. Pleiss, D. Blei, and J. Cunningham. Hierarchical Inducing Point Gaussian Process for Inter-domian Observations. In *Artificial Intelligence and Statistics*, 2021. Cited on page 7.

# A   Theory

**Preliminaries on Gaussian measures**

Since we are working in a setting beyond $\mathbb{R}^d$, we need a suitable notion of a multivariate Gaussian that can be employed in a coordinate-free manner. We employ the notion of a Gaussian in the sense of duality, given below. These notions are standard and classical, but since they are not well-known in machine learning, and for completeness, we prove the necessary properties ourselves.

**Definition 8.** *Let $(\Omega, \mathcal{F}, \mathbb{P})$ be a probability space. Let $V$ by a finite-dimensional real topological vector space, equipped with the standard topology, Borel $\sigma$-algebra, and the canonical pairing $\langle \cdot \mid \cdot \rangle : V^* \times V \to \mathbb{R}$ with its topological dual $V^*$. A random vector $v : \Omega \to V$ is called* Gaussian *if, for all $\phi \in V^*$, the random variable $\langle \phi \mid v \rangle : \Omega \to \mathbb{R}$ is univariate Gaussian.*

**Remark.**   It is not hard to show that in the setting of the definition above, the random variables $\langle \phi_1 \mid v \rangle, \dots, \langle \phi_k \mid v \rangle$ are jointly Gaussian for any finite collection $\phi_1, \dots, \phi_k \in V^*$. Indeed, this is equivalent to the Gaussianity of every linear combination $\alpha_1 \langle \phi_1 \mid v \rangle + \dots + \alpha_k \langle \phi_k \mid v \rangle = \langle \alpha_1 \phi_1 + \dots + \alpha_k \phi_k \mid v \rangle$, which is also ensured by the definition since $\alpha_1 \phi_1 + \dots + \alpha_k \phi_k \in V^*$.

We begin by showing that a Gaussian random vector in the sense of duality is characterized by a mean and a covariance, just like Gaussians in the standard, coordinate-dependent sense, starting with defining appropriate analogs of both notions in this setting.

**Lemma 9.** *For every Gaussian random vector $v$, there is a unique vector $\mu \in V$ and unique symmetric positive semi-definite bilinear form $k : V^* \times V^* \to \mathbb{R}$ such that for all $\phi \in V^*$, we have $\mathbb{E}\langle \phi \mid v \rangle = \langle \phi \mid \mu \rangle$ and $k(\phi, \psi) = \mathrm{Cov}(\langle \phi \mid v \rangle, \langle \psi \mid v \rangle)$. We say that $\mu$ is its* mean *and $k$ is its* covariance form*, and write $v \sim \mathrm{N}(\mu, k)$.*

*Proof.* Consider the map $\mathbb{E}\langle \cdot \mid v \rangle : V^* \to \mathbb{R}$. This map is a linear functional on the space $V^*$. Since $V$ is finite-dimensional, $V$ is reflexive, so there is exactly one vector $\mu \in V$ such that

$$\langle \phi \mid \mu \rangle = \mathbb{E}\langle \phi \mid v \rangle \tag{11}$$

for all $\phi \in V^*$. Next, define $k$ as

$$k(\phi, \psi) = \mathrm{Cov}(\langle \phi \mid v \rangle, \langle \psi \mid v \rangle) \tag{12}$$

for all $\phi, \psi \in V^*$. Clearly, $k$ is bilinear and positive semi-definite, that is $k(\phi, \phi) \geq 0$ for all $\phi \in V^*$. Thus the claim follows. $\qquad\square$

This tells us that every Gaussian random vector admits a mean and covariance: we now show that such Gaussians exist and are uniquely determined by this pair. Recall that for a measure $\pi$, and a measurable function $\phi$, the *pushforward measure* $\phi_* \pi$ is defined as $(\phi_* \pi)(A) = \pi(\phi^{-1}(A))$ for all measurable sets $A$.

**Lemma 10.** *For any vector $\mu \in V$ and any positive semi-definite bilinear form $k : V^* \times V^* \to \mathbb{R}$, there exists a random vector $v \sim \mathrm{N}(\mu, k)$. Moreover, if $w : \Omega \to V$ is another Gaussian random vector in the sense of Definition 8 with $w \sim \mathrm{N}(\mu, k)$, then $v$ and $w$ are identically distributed.*

*Proof.* Choose a basis $(e_i)$ on $V$, and let $(e^i)$ be the dual basis. Define the vector $\boldsymbol{\mu} \in \mathbb{R}^d$ and matrix $\mathbf{K} \in \mathbb{R}^{d \times d}$ by

$$\boldsymbol{\mu} = \begin{bmatrix} \langle e^1 \mid \mu \rangle \\ \vdots \\ \langle e^d \mid \mu \rangle \end{bmatrix} \qquad \mathbf{K} = \begin{bmatrix} k(e^1, e^1) & \dots & k(e^1, e^d) \\ \vdots & \ddots & \vdots \\ k(e^d, e^1) & \dots & k(e^d, e^d) \end{bmatrix}. \tag{13}$$

By positive semi-definiteness of $k$, the matrix $\mathbf{K}$ is a positive semi-definite matrix, so there exists a random vector $\boldsymbol{v} \sim \mathrm{N}(\boldsymbol{\mu}, \mathbf{K})$ in the classical Euclidean sense. Let $\mathcal{E} : V \to \mathbb{R}^n$ be the continuous linear isomorphism induced by the basis and define

$$v = \mathcal{E}^{-1} \boldsymbol{v}. \tag{14}$$

We claim that (a) $v$ is Gaussian, that is, if we test it against any covector, we obtain a univariate Gaussian, (b) the mean vector of $v$ is $\mu$, and (c) the covariance form of $v$ is $k$. To show (a), let $v^i$

denote the components of $v$ (scalar Gaussian random variables) so that $v = \sum_{i=1}^{d} v^i e_i$ and for any $\phi \in V^*$, write $\phi = \sum_{i=1}^{d} \phi_i e^i$, where $\phi_i = \langle \phi \mid e_i \rangle$. Then we have

$$\langle \phi \mid v \rangle = \left\langle \sum_{i=1}^{d} \phi_i e^i \;\middle|\; \sum_{j=1}^{d} v^j e_j \right\rangle = \sum_{i=1}^{d} \sum_{j=1}^{d} \phi_i v^j \underbrace{\langle e^i \mid e_j \rangle}_{\delta_{ij}} = \sum_{i=1}^{d} \phi_i v^i. \tag{15}$$

Since each $v^i$ is a univariate Gaussian, the linear combination on the right hand side is also a univariate Gaussian, which proves (a). To prove (b) and (c), we see that for any $\phi \in V^*$,

$$\mathbb{E}\langle \phi \mid v \rangle = \mathbb{E}\langle \phi \mid \sum_{i=1}^{d} v^i e_i \rangle = \mathbb{E} \sum_{i=1}^{d} v^i \langle \phi \mid e_i \rangle \tag{16}$$

$$= \sum_{i=1}^{d} \underbrace{(\mathbb{E}\, v^i)}_{\langle e^i \mid \mu \rangle} \langle \phi \mid e_i \rangle = \langle \phi \mid \sum_{i=1}^{d} \langle e^i \mid \mu \rangle e_i \rangle = \langle \phi \mid \mu \rangle. \tag{17}$$

Thus $v$ has the right mean. Now take an additional $\psi \in V^*$ and write

$$\mathrm{Cov}\Big(\langle \phi \mid v \rangle, \langle \psi \mid v \rangle\Big) = \mathbb{E}\big((\langle \phi \mid v \rangle - \langle \phi \mid \mu \rangle)(\langle \psi \mid v \rangle - \langle \psi \mid \mu \rangle)\big) \tag{18}$$

$$= \mathbb{E}\left(\sum_{i=1}^{d} (v^i - \langle e^i \mid \mu \rangle)\langle \phi \mid e_i \rangle\right)\left(\sum_{j=1}^{d} (v^j - \langle e^j \mid \mu \rangle)\langle \psi \mid e_j \rangle\right) \tag{19}$$

$$= \sum_{i=1}^{d} \sum_{j=1}^{d} \langle \phi \mid e_i \rangle \underbrace{\mathbb{E}\big((v^i - \langle e^i \mid \mu \rangle)(v^j - \langle e^j \mid \mu \rangle)\big)}_{k(e^i, e^j)} \langle \psi \mid e_j \rangle \tag{20}$$

$$= k\left(\sum_{i=1}^{d} \langle \phi \mid e_i \rangle e^i, \sum_{j=1}^{d} \langle \psi \mid e_j \rangle e^j\right) = k(\phi, \psi), \tag{21}$$

hence $v$ has the right covariance form.

Now let $w : \Omega \to V$ be another Gaussian random vector with $w \sim \mathrm{N}(\mu, k)$, and let $\pi_w$ be its pushforward measure. Similarly, let $\pi_v$ be the pushforward measure of $v$. Reversing the above argument, we see that pushforwards of measures $\pi_v$ and $\pi_w$ through $\mathcal{E}$, which we denote by $\pi_{\boldsymbol{v}}$ and $\pi_{\boldsymbol{w}}$, are both Gaussian distributions (in the classical sense) in $\mathbb{R}^d$ with the same mean vectors $\boldsymbol{\mu}$ and covariance matrices $\mathbf{K}$. Hence $\pi_{\boldsymbol{v}} = \pi_{\boldsymbol{w}}$ in distribution, but since $\mathcal{E}$ is a measurable space isomorphism,[6] we have $\pi_v = \pi_w$, which proves the claim. $\qquad \square$

Lemmas 9 and 10 show that a pair $\mu, k$ defines a unique probability distribution on $V$ which we call the Gaussian distribution with mean vector $\mu$ and covariance form $k$ on the vector space $V$ and denote by $\mathrm{N}(\mu, k)$. This establishes a notion of Gaussianity that is suitable and natural for describing finite-dimensional marginals in a coordinate-free manner.

**Existence and uniqueness (Proof of Theorem 4)**

Here, we prove that Gaussian vector fields exist and are uniquely determined by their mean vector field and cross-covariance kernel. Our goal now is, from a cross-covariance kernel, to construct a projective family of finite-dimensional marginals.

**Definition 11** (Preliminaries). *Let $X$ be a smooth manifold. Let*

$$\Gamma_{\mathrm{nns}}(TX) = \{f : X \to TX : \mathrm{proj}_X \circ f = \mathrm{id}_X\} \tag{22}$$

*be the vector space of not necessarily smooth sections.*

**Definition 12** (Cross-covariance kernel). *A symmetric function $k : T^*X \times T^*X \to \mathbb{R}$ is called fiberwise bilinear if at any pair of points $x, x' \in X$, we have*

$$k(\lambda \alpha_x + \mu \beta_x, \gamma_{x'}) = \lambda k(\alpha_x, \gamma_{x'}) + \mu k(\beta_x, \gamma_{x'}) \tag{23}$$

---

[6] A measurable space isomorphism is a measurable bijection with a measurable inverse.

*for any* $\alpha_x, \beta_x \in T_x^* X$, $\gamma_{x'} \in T_{x'}^* X$ *and* $\lambda, \mu \in \mathbb{R}$, *where we note by symmetry that the same requirement applies to its second argument. A fiberwise bilinear function $k$ is called* positive semi-definite *if for any set of covectors* $\alpha_{x_1}, \ldots, \alpha_{x_n} \in T^* X$, *we have*

$$\sum_{i=1}^{n} \sum_{j=1}^{n} k(\alpha_{x_i}, \alpha_{x_j}) \geq 0. \tag{24}$$

*We call a symmetric fiberwise bilinear positive semi-definite function a* cross-covariance kernel.

We show in the following example that this definition of the cross-covariance kernel is compatible with the notion of matrix-valued kernels used in classical vector-valued GPs and extends it naturally.

**Example 13** (Euclidean case)**.** *Consider $X = \mathbb{R}^d$ with a fixed inner product and an orthonormal basis, under which $\mathbb{R}^d$ is identified with $\left(\mathbb{R}^d\right)^*$. Consider a matrix-valued kernel $\kappa : \mathbb{R}^d \times \mathbb{R}^d \to \mathbb{R}^{d \times d}$ in the standard sense. Let $k((x, v), (x', v')) = v^T \kappa(x, x') v'$. Then $k : T^* \mathbb{R}^d \times T^* \mathbb{R}^d \to \mathbb{R}$ is a cross-covariance kernel in the above sense.*

*Indeed, $k$ is symmetric and fiberwise bilinear. Moreover, since $\kappa$ is positive semi-definite in the regular sense, we have that for arbitrary $x_1, \ldots, x_n \in \mathbb{R}^d$, the $nd \times nd$ matrix*

$$\Gamma(x_1, \ldots, x_n) = \begin{bmatrix} \kappa(x_1, x_1) & \ldots & \kappa(x_1, x_n) \\ \vdots & \ddots & \vdots \\ \kappa(x_n, x_1) & \ldots & \kappa(x_n, x_n) \end{bmatrix} \tag{25}$$

*is positive semi-definite, meaning that for an arbitrary collection $v_1, \ldots, v_n \in \mathbb{R}^d$, we have*

$$0 \leq \begin{bmatrix} v_1^T & \ldots & v_n^T \end{bmatrix} \begin{bmatrix} \kappa(x_1, x_1) & \ldots & \kappa(x_1, x_n) \\ \vdots & \ddots & \vdots \\ \kappa(x_n, x_1) & \ldots & \kappa(x_n, x_n) \end{bmatrix} \begin{bmatrix} v_1 \\ \vdots \\ v_n \end{bmatrix} = \sum_{i=1}^{n} \sum_{j=1}^{n} \underbrace{v_i^T \kappa(x_i, x_j) v_j}_{k((x_i, v_i), (x_j, v_j))}. \tag{26}$$

*Condition (24) thus follows, proving that this is a valid cross-covariance kernel.*

We proceed to introduce the system of coordinate-free finite-dimensional marginals that will be used to construct the vector-valued GP.

**Definition 14.** *Let $\mu \in \Gamma_{\mathrm{nns}}(TX)$ and $k : T^* X \times T^* X \to \mathbb{R}$ be a cross-covariance kernel. For any $x_1, \ldots, x_n \in X$, let $V_{x_1, \ldots, x_n} = T_{x_1} X \oplus \ldots \oplus T_{x_n} X$ and $V_{x_1, \ldots, x_n}^* = T_{x_1}^* X \oplus \ldots \oplus T_{x_n}^* X$. Define $\mu_{x_1, \ldots, x_n} \in V_{x_1, \ldots, x_n}$ and $k_{x_1, \ldots, x_n} : V_{x_1, \ldots, x_n}^* \times V_{x_1, \ldots, x_n}^* \to \mathbb{R}$ by*

$$\mu_{x_1, \ldots, x_n} = (\mu(x_1), \ldots, \mu(x_n)) \qquad k_{x_1, \ldots, x_n}(\alpha, \beta) = \sum_{i=1}^{n} \sum_{j=1}^{n} k(\alpha_{x_i}, \beta_{x_j}) \tag{27}$$

*for any $\alpha = (\alpha_{x_1}, \ldots, \alpha_{x_n})$, $\beta = (\beta_{x_1}, \ldots, \beta_{x_n}) \in V_{x_1, \ldots, x_n}^*$. We denote $\pi_{x_1, \ldots, x_n} = \mathrm{N}(\mu_{x_1, \ldots, x_n}, k_{x_1, \ldots, x_n})$ the system of marginals induced by $k$.*

We now prove existence and uniqueness of a measure on $\Gamma_{\mathrm{nns}}(TX)$ from the Gaussian measures defined on $V_{x_1, \ldots, x_n}$ for any $\{x_1, \ldots, x_n\} \subseteq X$. We do this by means of the general form of the Kolmogorov extension theorem formulated below. Recall again that for a measure $\pi$, and a measurable function $\phi$, the *pushfoward measure* $\phi_* \pi$ is defined as $(\phi_* \pi)(A) = \pi(\phi^{-1}(A))$ for all measurable sets $A$.

**Result 15** (Kolmogorov Extension Theorem)**.** *Let $(X_\alpha, \mathcal{B}_\alpha, \mathcal{O}_\alpha)_{\alpha \in A}$ be a family of measurable spaces, each equipped with a topology. For each finite $B \subseteq A$, let $\mu_B$ be an inner regular probability measure on $X_B = \prod_{\alpha \in B} X_\alpha$ with $\sigma$-algebra $\mathcal{B}_B$ and with the product topology $\mathcal{O}_B$ obeying*

$$(\mathrm{proj}_C)_* \mu_B = \mu_C \tag{28}$$

*whenever $C \subseteq B \subseteq A$ are two nested finite subsets of $A$. Here projections $\mathrm{proj}_C : X_B \to X_C$ are defined by $\mathrm{proj}_C(\{x_\alpha\}_{\alpha \in B}) = \{x_\alpha\}_{\alpha \in C}$ and $(\mathrm{proj}_C)_*$ denotes the pushforward by $\mathrm{proj}_C$. Then there exists a unique probability measure $\mu_A$ on $\mathcal{B}_A$ with the property that $(\mathrm{proj}_B)_* \mu_A = \mu_B$ for all finite $B \subseteq A$.*

*Proof.* Tao [42], Theorem 2.4.3. $\square$

By showing the existence of a probability measure on the space $\Gamma_{\text{nns}}(TX)$, one can start speaking about random variables $f : \Omega \to \Gamma_{\text{nns}}(TX)$ with said measure as their distribution: these are the Gaussian vector fields we seek. However, in order to apply the above result, we first need to verify condition (28). This is done in the following.

**Proposition 16.** *The family of measures* $(\pi_{x_1,\ldots,x_n})_{\{x_1,\ldots,x_n\}\subseteq X}$ *is a projective family in the sense that for any* $\{x_1,\ldots,x_m\} \subseteq \{x_1,\ldots,x_n\} \subseteq X$, *we have*

$$(\text{proj}_{x_1,\ldots,x_m})_* \pi_{x_1,\ldots,x_n} = \pi_{x_1,\ldots,x_m} \tag{29}$$

*where* $\text{proj}_{x_1,\ldots,x_m} : V_{x_1,\ldots,x_n} \to V_{x_1,\ldots,x_m}$ *is the canonical projection induced by the direct sum.*

*Proof.* Take two random variables $v_{x_1,\ldots,x_n} : \Omega \to V_{x_1,\ldots,x_n}$ and $v_{x_1,\ldots,x_m} : \Omega \to V_{x_1,\ldots,x_m}$ with $v_{x_1,\ldots,x_n} \sim \pi_{x_1,\ldots,x_n}$ and $v_{x_1,\ldots,x_m} \sim \pi_{x_1,\ldots,x_m}$. It suffices to show that for the random variable $v_{x_1,\ldots,x_n} : \Omega \to V_{x_1,\ldots,x_m}$ we have

$$v_{x_1,\ldots,x_m} \overset{\text{d}}{=} \text{proj}_{x_1,\ldots,x_m} v_{x_1,\ldots,x_n} \tag{30}$$

where $\overset{\text{d}}{=}$ denotes the equality of distributions. We first show that $\text{proj}_{x_1,\ldots,x_m} v_{x_1,\ldots,x_n}$ is Gaussian. Let $\phi \in V^*_{x_1,\ldots,x_m}$ and write

$$\big\langle \phi \mid \text{proj}_{x_1,\ldots,x_m} v_{x_1,\ldots,x_n} \big\rangle = \big\langle (\phi,0) \mid v_{x_1,\ldots,x_n} \big\rangle \tag{31}$$

where $(\phi,0) \in V^*_{x_1,\ldots,x_n}$ is the natural inclusion of $\phi \in V^*_{x_1,\ldots,x_m}$ in the space $V^*_{x_1,\ldots,x_n}$ by padding with the zero vector over all components of the direct sum whose indices are not $x_1,\ldots,x_m$. This identity holds for all vectors, hence it holds for random vectors, and $\text{proj}_{x_1,\ldots,x_m} v_{x_1,\ldots,x_n}$ is Gaussian. Now, we compute its moments: write

$$\mathbb{E}\big\langle \phi \mid \text{proj}_{x_1,\ldots,x_m} v_{x_1,\ldots,x_n} \big\rangle = \mathbb{E}\big\langle (\phi,0) \mid v_{x_1,\ldots,x_n} \big\rangle \tag{32}$$
$$= \big\langle (\phi,0) \mid \mu_{x_1,\ldots,x_n} \big\rangle \tag{33}$$
$$= \big\langle \phi \mid \text{proj}_{x_1,\ldots,x_m} \mu_{x_1,\ldots,x_n} \big\rangle \tag{34}$$
$$= \big\langle \phi \mid \mu_{x_1,\ldots,x_m} \big\rangle \tag{35}$$

where the last line follows by definition of $\mu_{x_1,\ldots,x_m}$, and

$$\text{Cov}(\big\langle \phi \mid \text{proj}_{x_1,\ldots,x_m} v_{x_1,\ldots,x_n} \big\rangle, \big\langle \psi \mid \text{proj}_{x_1,\ldots,x_m} v_{x_1,\ldots,x_n} \big\rangle) \tag{36}$$
$$= \text{Cov}\left(\big\langle (\phi,0) \mid v_{x_1,\ldots,x_n} \big\rangle, \big\langle (\psi,0) \mid v_{x_1,\ldots,x_n} \big\rangle\right) \tag{37}$$
$$= k_{x_1,\ldots,x_n}((\phi,0),(\psi,0)) \tag{38}$$
$$= k_{x_1,\ldots,x_m}(\phi,\psi) \tag{39}$$

where the last line follows by bilinearity and the definition of $k_{x_1,\ldots,x_m}$.

So far we have shown that $\text{proj}_{x_1,\ldots,x_m} v_{x_1,\ldots,x_n}$ is Gaussian over $V_{x_1,\ldots,x_m}$ and its mean vector and covariance form coincide with those of $v_{x_1,\ldots,x_m}$. Hence, by the uniqueness part of Lemma 10 we have $v_{x_1,\ldots,x_m} \overset{d}{=} \text{proj}_{x_1,\ldots,x_m} v_{x_1,\ldots,x_n}$. This finishes the proof. $\square$

We are now ready to apply the Kolmogorov extension theorem to show existence of the desired distribution.

**Proposition 17.** *There exists a unique measure* $\pi_\infty$ *on the infinite product space* $\prod_{x\in X} T_x X$.[7]

*Proof.* We apply the prior result 15. Let $X$ be the index set, and take $(T_x X)_{x\in X}$, equipped with the standard topology and Borel $\sigma$-algebra as our measurable spaces. For each finite $\{x_1,\ldots,x_n\} \subseteq X$, take $\pi_{x_1,\ldots,x_n}$ as our probability measure, and note that since each $\pi_{x_1,\ldots,x_n}$ is a finite measure on a finite-dimensional real vector space $V_{x_1,\ldots,x_n}$, it is automatically inner regular. Moreover, the family of measures $(\pi_{x_1,\ldots,x_n})_{\{x_1,\ldots,x_n\}\subseteq X}$ is projective by Proposition 16. The claim follows. $\square$

This gives our GP as a measure on an infinite Cartesian space: we now map this measure into the space of sections.

---

[7] Note that this is the Tychonoff product of topological spaces rather than a direct product of linear spaces.

**Corollary 18.** *There exists a unique measure $\pi_{\Gamma_{\mathrm{nns}}(TX)}$ on $\Gamma_{\mathrm{nns}}(TX)$ equipped with the pushforward $\sigma$-algebra.*

*Proof.* Define the operator $\mathcal{I} : \prod_{x \in X} T_x X \to \Gamma_{\mathrm{nns}}(TX)$ by

$$(\mathcal{I}s)(x) = (x, \mathrm{proj}_x s) \tag{40}$$

for all $x \in X$ and $s \in \prod_{x \in X} T_x X$. Take $\pi_{\Gamma_{\mathrm{nns}}(TX)} = \mathcal{I}_* \pi_\infty$. $\qquad\square$

This is the probability distribution of our Gaussian process. We are now ready to define Gaussian vector fields, and show that each Gaussian vector field in turn possesses a mean vector field and cross-covariance kernel.

**Definition 19.** *Let $X$ be a manifold. We say that a random vector field $f : \Omega \to \Gamma_{\mathrm{nns}}(TX)$ is Gaussian if for any finite set of locations $(x_1, \ldots, x_n) \in X^n$, the random vector $f(x_1), \ldots, f(x_n) \in T_{x_1} X \oplus \ldots \oplus T_{x_n} X$ is Gaussian in the sense of Definition 8.*

**Definition 20.** *Let $f : \Omega \to \Gamma_{\mathrm{nns}}(TX)$ be a Gaussian vector field. Define $\mu$ to be the unique vector field for which, for any $x \in X$ and any $\phi \in T_x^*$, we have that*

$$\langle \phi \mid \mu(x) \rangle = \mathbb{E}\langle \phi \mid f(x) \rangle. \tag{41}$$

*Next taking an additional $x' \in X$ and $\psi \in T_{x'}^*$, define the cross-covariance kernel $k$ by*

$$k(\phi, \psi) = \mathrm{Cov}(\langle \phi \mid f(x) \rangle, \langle \psi \mid f(x') \rangle). \tag{42}$$

Summarizing, we obtain the following claim.

**Theorem 21.** *Every pair consisting of a mean vector field and symmetric fiberwise bilinear positive definite function $k : T^* X \times T^* X \to \mathbb{R}$, which we call a cross-covariance kernel, defines a unique (distribution-wise) Gaussian vector field in the sense of Definition 19. Conversely, every Gaussian vector field admits and is characterized uniquely by this pair.*

*Proof.* Corollary 18, Definition 19, and Definition 20. $\qquad\square$

### Embeddings (Proof of Proposition 2)

**Proposition 22.** *Let $\mathrm{emb} : X \to \mathbb{R}^p$ be an embedding, let $f$ be a Gaussian vector field on $X$, and denote by $\boldsymbol{f}_{\mathrm{emb}} : \mathrm{emb}(X) \to \mathbb{R}^p$ its pushforward along the embedding, that is, for any $x \in X$,*

$$\boldsymbol{f}_{\mathrm{emb}}(\mathrm{emb}(x)) = \mathrm{d}_x\mathrm{emb}(f(x)), \tag{43}$$

*where $\mathrm{d}_x\mathrm{emb} : T_x X \to T_{\mathrm{emb}(x)}\mathbb{R}^p$ is the differential of $\mathrm{emb}$. Then $\boldsymbol{f}_{\mathrm{emb}}$ is a vector-valued Gaussian process in the standard sense.*

*Proof.* Let $x_1, \ldots, x_n \in X^n$ be a finite set of arbitrary locations. In what follows, we use a slight abuse of notation by letting $x_i$ denote both $x_i$ and $\mathrm{emb}(x_i)$ for simplicity. We claim that the random vector $(\boldsymbol{f}_{\mathrm{emb}}(x_1), \ldots, \boldsymbol{f}_{\mathrm{emb}}(x_n)) \in \mathbb{R}^{np}$ is multivariate Gaussian, which is sufficient to prove our result. Since $f$ is a Gaussian vector field, we have that

$$(f(x_1), \ldots, f(x_n)) \sim \mathrm{N}(\mu_{x_1,\ldots,x_n}, k_{x_1,\ldots,x_n}) \tag{44}$$

is a Gaussian random vector on $T_{x_1} X \oplus \ldots \oplus T_{x_n} X$. Now consider the map $\phi_{x_1,\ldots,x_n} : T_{x_1} X \oplus \ldots \oplus T_{x_n} X \to T_{\mathrm{emb}(x_1)}\mathbb{R}^p \oplus \ldots \oplus T_{\mathrm{emb}(x_n)}\mathbb{R}^p \cong \mathbb{R}^{np}$ defined as

$$\phi_{x_1,\ldots,x_n}(f_{x_1}, \ldots, f_{x_n}) = (\mathrm{d}_{x_1}\mathrm{emb}(f_{x_1}), \ldots, \mathrm{d}_{x_n}\mathrm{emb}(f_{x_n})), \tag{45}$$

for all $(f_{x_1}, \ldots, f_{x_n}) \in T_{x_1} X \oplus \ldots \oplus T_{x_n} X$, which is linear, owing to the linearity of $\mathrm{d}_x\mathrm{emb}$. Since linear maps preserve Gaussianity, it follows that the vector $\phi_{x_1,\ldots,x_n}(f(x_1), \ldots, f(x_n)) = (\boldsymbol{f}_{\mathrm{emb}}(x_1), \ldots, \boldsymbol{f}_{\mathrm{emb}}(x_n)) \in \mathbb{R}^{np}$ is multivariate Gaussian and the claim follows. $\qquad\square$

## Coordinate Expressions (Proof of Proposition 5)

We recall the definition of a *frame* on $X$ and its dual object, namely, the *coframe*.

**Definition 23.** *A* frame $F$ *on* $X$ *is defined as a collection* $(e_i)_{i=1}^d$ *of not necessarily smooth sections of* $TX$ *such that at each point* $x \in X$, *the vectors* $(e_i(x))_{i=1}^d$ *form a basis of* $T_xX$. *The corresponding* coframe $F^*$ *is defined as a collection* $(e^i)_{i=1}^d$ *of not necessarily smooth sections of* $T^*X$ *such that* $\langle e^i(x) | e_j(x) \rangle = \delta_{ij}$ *for all* $x \in X$.

**Proposition 24.** *Let* $f : \Omega \to \Gamma_{\mathrm{nns}}(TX)$ *be a Gaussian vector field on* $X$ *with cross-covariance kernel* $k : T^*X \times T^*X \to \mathbb{R}$. *Given a frame* $F = (e_1, \ldots, e_d)$ *on* $X$ *and* $F^* = (e^1, \ldots, e^d)$ *be its coframe, define* $f^i = \langle e^i \mid f \rangle$ *for all* $i = 1, \ldots, d$. *Then* $\boldsymbol{f} = (f^1, \ldots, f^d) : \Omega \times X \to \mathbb{R}^d$ *is a vector-valued GP in the usual sense with matrix-valued kernel* $\mathbf{K}_F : X \times X \to \mathbb{R}^{d \times d}$ *given by*

$$\mathbf{K}_F(x, x') = \begin{bmatrix} k(e^1(x), e^1(x')) & \ldots & k(e^1(x), e^d(x')) \\ \vdots & \ddots & \vdots \\ k(e^d(x), e^1(x')) & \ldots & k(e^d(x), e^d(x')) \end{bmatrix}. \tag{46}$$

*Conversely, given a vector-valued GP* $\boldsymbol{f} = (f^1, \ldots, f^d) : \Omega \times X \to \mathbb{R}^d$ *and a frame* $F = (e_1, \ldots, e_d)$ *on* $X$, $f(\cdot) := \sum_{i=1}^d f^i(\cdot)e_i(\cdot)$ *defines a Gaussian vector field on* $X$.

*Proof.* First, we note that $f^i(x) = \langle e^i(x) \mid f(x) \rangle$ are jointly Gaussian for all $i = 1, \ldots, d$ and all $x \in X$. Thus for any $x_1, \ldots, x_n \in X$, the vector $(\boldsymbol{f}(x_1), \ldots, \boldsymbol{f}(x_n)) \in \mathbb{R}^{n \times d}$ is multivariate Gaussian and therefore $\boldsymbol{f}$ is a vector-valued GP in the usual sense. Now for any $x, x' \in X$, the kernel of $\boldsymbol{f}$ evaluated at these points reads

$$\mathbf{K}_F(x, x') = \begin{bmatrix} \mathrm{Cov}(f^1(x), f^1(x')) & \ldots & \mathrm{Cov}(f^1(x), f^d(x')) \\ \vdots & \ddots & \vdots \\ \mathrm{Cov}(f^d(x), f^1(x')) & \ldots & \mathrm{Cov}(f^d(x), f^d(x')) \end{bmatrix} \tag{47}$$

$$= \begin{bmatrix} k(e^1(x), e^1(x')) & \ldots & k(e^1(x), e^d(x')) \\ \vdots & \ddots & \vdots \\ k(e^d(x), e^1(x')) & \ldots & k(e^d(x), e^d(x')) \end{bmatrix}, \tag{48}$$

which follows from Definition 20. This concludes the first part of the proof.

To prove the converse direction, for any collection of points $x_1, \ldots, x_n \in X$, define the random vector $v_{x_1, \ldots, x_n} = (f(x_1), \ldots, f(x_n))$, where $f$ is given by $f(x) = \sum_{i=1}^d f^i(x)e_i(x)$. Now for any $\phi_{x_1, \ldots, x_n} = (\phi_{x_1}, \ldots, \phi_{x_n}) \in V^*_{x_1, \ldots, x_n}$, we have

$$\langle \phi_{x_1, \ldots, x_n} \mid v_{x_1, \ldots, x_n} \rangle = \sum_{i=1}^n \langle \phi_{x_i} \mid f(x_i) \rangle \tag{49}$$

$$= \sum_{i=1}^n \langle \phi_{x_i} \mid \sum_{j=1}^d f^j(x_i)e_j(x_i) \rangle \tag{50}$$

$$= \sum_{i=1}^n \sum_{j=1}^d f^j(x_i)\langle \phi_{x_i} \mid e_j(x_i) \rangle. \tag{51}$$

Since $f^j(x_i)$ is univariate Gaussian for all $i = 1, \ldots, n$ and $j = 1, \ldots, d$, the above linear combination is univariate Gaussian and therefore $v_{x_1, \ldots, x_n}$ is Gaussian in the sense of Definition 8. Since $x_1, \ldots, x_n$ were chosen arbitrarily, $f$ is a Gaussian vector field. $\square$

## Gauge Independence (Proof of Corollary 6)

Given two frames $F, F'$ on $X$, an abstract vector $f_x \in T_xX$ has two vector representations $\boldsymbol{f}_x, \boldsymbol{f}'_x$ in the respective frames. Recall that $F'$ is said to be obtained from $F$ by a *gauge transformation* with respect to a matrix field $\mathbf{A} : X \to \mathrm{GL}(d, \mathbb{R})$, if

$$\boldsymbol{f}'_x = \mathbf{A}(x)\boldsymbol{f}_x \tag{52}$$

holds for all $x \in X$, and we write $F' = \mathbf{A}F$. In the following, we compute an explicit expression for the gauge-transformed frame $\mathbf{A}F$ and its coframe.

**Lemma 25.** *Let $F = (e_1, \ldots, e_d)$ be a frame on $X$, $\mathbf{A} : X \to \mathrm{GL}(d, \mathbb{R})$ be a matrix field of gauge transformations, $\mathbf{A}F = (\varepsilon_1, \ldots, \varepsilon_d)$ be the gauge transformed frame as above and let $(\mathbf{A}F)^* = (\varepsilon^1, \ldots, \varepsilon^d)$ be the corresponding coframe. Then we have the following explicit expressions*

$$\varepsilon_i(x) = \sum_{j=1}^{d} e_j(x)[\mathbf{A}^{-1}(x)]_{ji}, \qquad\qquad \varepsilon^i(x) = \sum_{j=1}^{d} [\mathbf{A}(x)]_{ij} e^j(x). \tag{53}$$

*Proof.* For any $x \in X$, let $f_x \in T_x X$ be an abstract vector, which has the vector representations $\boldsymbol{f}_x$ and $\mathbf{A}(x)\boldsymbol{f}_x$ in the frames $F$ and $\mathbf{A}F$ respectively. Letting $\boldsymbol{f}_x = (f_x^1, \ldots, f_x^d)$, we have

$$f_x = \sum_{i=1}^{d} f_x^i e_i(x) = \sum_{i=1}^{d} \sum_{j=1}^{d} ([\mathbf{A}(x)]_{ji} f_x^i) \varepsilon_j(x) = \sum_{i=1}^{d} f_x^i \left( \sum_{j=1}^{d} \varepsilon_j(x)[\mathbf{A}(x)]_{ji} \right). \tag{54}$$

Thus, $e_i(x) = \sum_{j=1}^{d} \varepsilon_j(x)[\mathbf{A}(x)]_{ji}$, or identically, $\varepsilon_i(x) = \sum_{j=1}^{d} e_j(x)[\mathbf{A}^{-1}(x)]_{ji}$. We now claim that $\varepsilon^i(x) = \sum_{j=1}^{d} [\mathbf{A}(x)]_{ij} e^j(x)$, which we prove by showing that it satisfies the relation $\langle \varepsilon^i(x) \mid \varepsilon_j(x) \rangle = \delta_{ij}$ as follows:

$$\langle \varepsilon^i(x) \mid \varepsilon_j(x) \rangle = \left\langle \sum_{k=1}^{d} [\mathbf{A}(x)]_{ik} e^k(x) \mid \sum_{l=1}^{d} e_l(x)[\mathbf{A}^{-1}(x)]_{lj} \right\rangle \tag{55}$$

$$= \sum_{k=1}^{d} \sum_{l=1}^{d} [\mathbf{A}(x)]_{ik} \underbrace{\langle e^k(x) \mid e_l(x) \rangle}_{\delta_{kl}} [\mathbf{A}^{-1}(x)]_{lj} \tag{56}$$

$$= \sum_{k=1}^{d} [\mathbf{A}(x)]_{ik} [\mathbf{A}^{-1}(x)]_{kj} \tag{57}$$

$$= \underbrace{[\mathbf{A}(x)\mathbf{A}^{-1}(x)]_{ij}}_{\delta_{ij}}. \tag{58}$$

This concludes the proof. $\qquad\square$

The following is, then, straightforward to show.

**Corollary 26.** *Let $F$ be a frame on $X$ and $\mathbf{K}_F : X \times X \to \mathbb{R}^{d \times d}$ be the corresponding matrix representation of a cross-covariance kernel $k : T^*X \times T^*X \to \mathbb{R}$. This satisfies the* equivariance *condition*

$$\mathbf{K}_{\mathbf{A}F}(x, x') = \mathbf{A}(x)\mathbf{K}_F(x, x')\mathbf{A}(x')^T, \tag{59}$$

*where $\mathbf{A} : X \to \mathrm{GL}(d, \mathbb{R})$ is a gauge transformation applied to each point on $X$. All cross-covariance kernels in the sense of Proposition 4 arise this way.*

*Proof.* Let $F = (e_1, \ldots, e_d)$ and $\mathbf{A}F = (\varepsilon_1, \ldots, \varepsilon_d)$. Then by the previous lemma, we have

$$[\mathbf{K}_{\mathbf{A}F}(x, x')]_{ij} = k(\varepsilon^i(x), \varepsilon^j(x')) \tag{60}$$

$$= \sum_{k=1}^{d} \sum_{l=1}^{d} k\left( [\mathbf{A}(x)]_{ik} e^k(x), [\mathbf{A}(x')]_{jl} e^l(x') \right) \tag{61}$$

$$= \sum_{k=1}^{d} \sum_{l=1}^{d} [\mathbf{A}(x)]_{ik} [\mathbf{K}_F(x, x')]_{kl} [\mathbf{A}(x')]_{jl}, \tag{62}$$

which proves the identity (59).

The second claim is obvious: take some cross-covariance kernel in the sense of Proposition 4 and some frame—this induces a gauge independent matrix-valued kernel that correspond to the cross-covariance kernel in the sense of Proposition 4 from which it was constructed in the first place. $\qquad\square$

**Projected Kernels (Proof of Proposition 7)**

Here we formally describe the projected kernel construction. We start by noting some properties of the projection matrices associated with differentials of isometric embeddings.

**Lemma 27.** *Let $(X, g)$ be a Riemannian manifold and* $\mathrm{emb} : X \to \mathbb{R}^{d'}$ *be an isometric embedding. Given a frame $F = (e_1, \ldots, e_d)$ on $X$, denote by $\mathbf{P}_{(\cdot)} : X \to \mathbb{R}^{d \times d'}$ its associated projection matrix, defined for every $x$ as the matrix representation of $\mathrm{d}_x\mathrm{emb}$ within $F$, and $\mathbf{\Gamma} : X \to \mathbb{R}^{d \times d}$, the matrix field representation of the Riemannian metric $g$, that is, $[\mathbf{\Gamma}(x)]_{ij} = g_x(e_i(x), e_j(x))$ for all $i, j = 1, \ldots, d$ and $x \in X$. Then we have*

$$\mathbf{P}_x \mathbf{P}_x^T = \mathbf{\Gamma}(x). \tag{63}$$

*Proof.* Since the embedding is isometric, for any $v, v' \in T_x X$, we have

$$g_x(u, v) = \langle \mathrm{d}_x\mathrm{emb}(v), \mathrm{d}_x\mathrm{emb}(v') \rangle, \tag{64}$$

which, in the corresponding vector representation with respect to a frame $F$, reads

$$\boldsymbol{v}^T \mathbf{\Gamma}(x) \boldsymbol{v}' = \langle \mathbf{P}_x^T \boldsymbol{v}, \mathbf{P}_x^T \boldsymbol{v}' \rangle = \boldsymbol{v}^T (\mathbf{P}_x \mathbf{P}_x^T) \boldsymbol{v}'. \tag{65}$$

for any $\boldsymbol{v}, \boldsymbol{v}'$. This implies that $\mathbf{\Gamma}(x) = \mathbf{P}_x \mathbf{P}_x^T$ for all $x$ and proves the claim. $\qquad\square$

We proceed to describe the projected kernel construction, which lets us transform a matrix-valued kernel on an ambient space into a cross-covariance kernel on the manifold.

**Proposition 28.** *Let $(X, g)$ be a Riemannian manifold,* $\mathrm{emb} : X \to \mathbb{R}^{d'}$ *be an isometric embedding and $F$ be a frame on $X$. We denote by $\mathbf{P}_{(\cdot)} : X \to \mathbb{R}^{d \times d'}$ the associated projection matrix under $F$, and let $\boldsymbol{f}' : X \to \mathbb{R}^{d'}$ be any vector-valued Gaussian process with matrix-valued kernel $\boldsymbol{\kappa} : X \times X \to \mathbb{R}^{d' \times d'}$. Then, the vector-valued function $\boldsymbol{f} = \mathbf{P}\boldsymbol{f}' : X \to \mathbb{R}^d$ defines a Gaussian vector field $f$ on $X$ using the construction in Proposition 24, whose kernel under the frame $F$ has matrix representation*

$$\mathbf{K}_F(x, x') = \mathbf{P}_x \boldsymbol{\kappa}(x, x') \mathbf{P}_{x'}^T. \tag{66}$$

*Moreover, all cross-covariance kernels $k : T^*X \times T^*X \to \mathbb{R}$ arise this way.*

*Proof.* We demonstrate the first part by computing the covariance of $\boldsymbol{f}$. For any $x, x' \in X$, we have

$$\mathbf{K}_F(x, x')_{ij} = \mathrm{Cov}(f^i(x), f^j(x')) \tag{67}$$

$$= \mathrm{Cov}([\mathbf{P}_x \boldsymbol{f}'(x)]_i, [\mathbf{P}_{x'} \boldsymbol{f}'(x')]_j) \tag{68}$$

$$= \sum_{k=1}^d \sum_{l=1}^d \mathrm{Cov}([\mathbf{P}_x]_{ik} f_k'(x), [\mathbf{P}_{x'}]_{jl} f_l'(x')) \tag{69}$$

$$= \sum_{k=1}^d \sum_{l=1}^d [\mathbf{P}_x]_{ik} \mathrm{Cov}(f_k'(x), f_l'(x')) [\mathbf{P}_{x'}]_{jl} \tag{70}$$

$$= \sum_{k=1}^d \sum_{l=1}^d [\mathbf{P}_x]_{ik} \boldsymbol{\kappa}(x, x')_{kl} [\mathbf{P}_{x'}]_{jl}, \tag{71}$$

which proves the identity (66).

Conversely, let $k : T^*X \times T^*X \to \mathbb{R}$ be a cross-covariance kernel. We first construct a matrix-valued kernel $\mathbf{K}_F$ as in Proposition 24. Define

$$\boldsymbol{\kappa}(x, x') = \mathbf{P}_x^T \mathbf{K}_{\mathbf{\Gamma}^{-1}F}(x, x') \mathbf{P}_{x'}, \tag{72}$$

where $\mathbf{\Gamma} : X \to \mathbb{R}^{d \times d}$ is the matrix field representation of the metric $g$ as given in the statement of Lemma 27. Then by the same lemma, we have

$$\mathbf{P}_x \boldsymbol{\kappa}(x, x') \mathbf{P}_{x'}^T = (\mathbf{P}_x \mathbf{P}_x^T) \mathbf{K}_{\mathbf{\Gamma}^{-1}F}(x, x') (\mathbf{P}_{x'} \mathbf{P}_{x'}^T) \tag{73}$$

$$= \mathbf{\Gamma}(x) \mathbf{K}_{\mathbf{\Gamma}^{-1}F}(x, x') \mathbf{\Gamma}(x') \tag{74}$$

$$= \mathbf{K}_F(x, x'), \tag{75}$$

where we used that $\mathbf{\Gamma}(x)^T = \mathbf{\Gamma}(x)$ and Corollary 26 to deduce the last equality. Thus, any cross-covariance kernel $k$ can be obtained from a matrix-valued kernel $\boldsymbol{\kappa}$ on the ambient space and therefore we do not lose any generality by working with the latter. $\qquad\square$

## B  Experimental details

Here, we include further details about the experiments conducted in Section 4. All experiments were conducted on a single workstation with 64GB RAM, using CPU-based computation.

**Fourier features for product kernels**

Throughout this paper we use the sparse GP formulation of Wilson et al. [47, 48] to work with GPs. In order to apply this method we need to be able to sample a Fourier feature approximation of the kernel. For stationary kernels supported on Euclidean space one typically uses a random Fourier feature (RFF) approximation [33]

$$\widetilde{f}(\cdot) = \frac{1}{\sqrt{l}} \sum_{i=1}^{l} w_i \phi_i(x), \qquad\qquad w_i \sim \mathrm{N}(0,1), \qquad (76)$$

where the $\phi_i$ are Fourier basis functions sampled from the spectral density of the kernel—see Sutherland and Schneider [41] for details. The resulting random function $\widetilde{f}(\cdot)$ is then a Gaussian process with zero mean and kernel $l^{-1}\boldsymbol{\Phi}(\cdot)^T\boldsymbol{\Phi}(\cdot)$, where $\boldsymbol{\Phi}$ is a vector of the $l$ basis functions. This approximates the true GP with a dimension-free error of the order $l^{-1/2}$.

For kernels supported on compact spaces we use a Karhunen–Loéve (KL) expansion. If we have a Gaussian process $f(\cdot)$ on a compact space, then we can optimally approximate this function (in terms of $L^2$-norm) by truncating its KL expansion

$$f(\cdot) = \sum_{i=1}^{\infty} w_i \psi_i(x) \qquad\qquad w_i \sim \mathrm{N}(0, \lambda_i) \qquad (77)$$

where $\phi_i, \lambda_i$ are the $i^{\text{th}}$ eigenfunctions and values of the kernel, $\int_X \psi(x)k(x,\cdot)\,\mathrm{d}x = \lambda_i\psi_i(\cdot)$, sorted in descending order of the eingenvalues. For the squared exponential and Matérn kernels on compact manifolds, these eigenfunctions are the eigenfunctions of the Laplacian of the manifold, and the eigenvalues are given by a transformation of the Laplacian eigenvalues [2].

The question then arises of what to do in the case of a product of kernels, each taking as input some different space, where some are suited to RFF approximation, and some to a KL approximation. We propose the following approach.

1. All the RFF-appropriate kernels can be combined into one approximation by sampling the basis functions from the product measure of their Fourier transforms.

2. All of the KL-appropriate kernels can be combined into one approximation by computing the $k$ largest eigenvalues of the product manifold the kernels are defined on. If we have two compact manifolds with eigenvalue-function pairs $(\alpha_i, f_i(\cdot))_{i=1}^{\infty}$ and $(\beta_j, g_j(\cdot))_{j=1}^{\infty}$, then the eigenvalue-function pairs on the product manifold are $(\alpha_i + \beta_j, f_i(\cdot)g_j(\cdot))_{i,j=1}^{\infty,\infty}$ [4]. We can repeatedly apply this to find the approximation for the kernel on arbitrary products of compact manifolds.

3. Define the Fourier feature approximation of the combination of this RFF and KL approximations as

$$f(e, m) = \frac{1}{\sqrt{l}} \sum_{i=1}^{l} \sum_{j=1}^{k} w_{i,j} \phi_i(e) \psi_j(m) \qquad w_{i,j} \sim \mathcal{N}(0, \lambda_j) \qquad (78)$$

where $e, m$ are the inputs to the RFF and KL appropriate kernels respectively, $\phi_i$ are the basis functions of the Euclidean kernels sampled from the product measure, and $\lambda_j, \psi_j$, are the product eigenpairs on the product manifold. In the limit of infinite basis functions in both $l$ and $k$ this will give the correct kernel, and therefore the true prior.

**Dynamics experiment**

In this experiment, the base manifold is the state space of the single pendulum system. The position of the pendulum is represented by a single angle in $[0, 2\pi)$, which corresponds to the circle $\mathbb{S}^1$. The

momentum lies then in its respective cotangent space. The phase space is the product of these, $\mathbb{S}^1 \times \mathbb{R}^1$.

This product manifold naturally embeds into $\mathbb{R}^3$ by embedding the circle into $\mathbb{R}^2$ in the canonical way, and leaving $\mathbb{R}^1$ unchanged. The embedding is then

$$\text{emb}(q, p) = (\cos q, \sin q, p), \tag{79}$$

where $q$ is the position and $p$ is the angular momentum. The global projection matrix given by

$$\mathbf{P}_{q,p} = \begin{bmatrix} -\sin q & \cos q & 0 \\ 0 & 0 & 1 \end{bmatrix}. \tag{80}$$

The Euclidean vector kernel we use is a separable kernel, produced by taking the product of an intrinsic squared exponential manifold kernel with the identity matrix to give a matrix valued kernel, $\boldsymbol{\kappa} = k_{\mathbb{S}^1 \times \mathbb{R}^1} \mathbf{I}_{3 \times 3}$. The intrinsic manifold kernel is produced by the product of a typical Euclidean squared exponential kernel with a squared exponential kernel defined on $\mathbb{S}^1$ by Borovitskiy et al. [2], so that $k_{\mathbb{S}^1 \times \mathbb{R}^1} = k_{\mathbb{S}^1} k_{\mathbb{R}^1}$. The length scales of these kernels are set to 0.3 and 1.2 respectively, and the amplitude set to give $k(x, x) = 1$.

To learn the dynamics, we initialise the system at two start points, and evolve the system using leapfrog integration. From these observations of position, we backward Euler integrate the momentum of the system, $p_i = \frac{1}{2}ml(q_{i+1} - q_i)$, and from these position-momentum trajectories we estimate observations of the dynamics field

$$\nabla_t(q, p)_i = \left( \frac{q_{i+1} - q_i}{h}, \frac{p_{i+1} - p_i}{h} \right) \tag{81}$$

where $h = 0.01$ is the step size. Using these observations, we condition a sparse GP using all the data using the analytic expression for the sparse posterior kernel matrix. The result is an estimate of the system dynamics with suitable uncertainty estimates. In order to compute rollouts of these dynamics, we follow Wilson et al. [47, 48] and employ linear-time pathwise sampling of this sparse GP together with leapfrog integration [16].

**Wind interpolation experiment**

In this experiment, the base manifold is the sphere $\mathbb{S}^2$, which we embed naturally in $\mathbb{R}^3$ as

$$\text{emb}(\phi, \theta) = (\cos \theta \sin \phi, \sin \theta \sin \phi, \cos \phi), \tag{82}$$

where we used spherical coordinates $\phi \in (0, \pi), \theta \in [0, 2\pi)$ to parametrise the sphere

$$(\phi, \theta) \in \{(0, 0)\} \cup \{(\pi, 0)\} \cup (0, \pi) \times [0, 2\pi) \tag{83}$$

We choose a frame $F = (e_1, e_2)$, where $e_1(\phi, \theta) = \hat{\phi}$ and $e_2(\phi, \theta) = \hat{\theta}$ are the unit vectors in the $\phi, \theta$ directions respectively for all $\phi \in (0, \pi), \theta \in (0, 2\pi)$. The choice of points on the North and South poles determines the choice of gauge at these points. The corresponding orthonormal projection matrix reads

$$\mathbf{P}_{\phi,\theta} = \begin{bmatrix} \cos \theta \cos \phi & \sin \theta \cos \phi & -\sin \phi \\ -\sin \theta & \cos \theta & 0 \end{bmatrix}, \tag{84}$$

for all points, with the choice of $\theta = 0$ giving the choice of frame at the poles.

For the data, we used the following publicly available data sets.

- The ERA5 atmospheric reanalysis data. In particular, the variables 10M-U-COMPONENT-OF-WIND and 10M-V-COMPONENT-OF-WIND from the REANALYSIS-ERA5-SINGLE-LEVELS dataset for the date 01/01/2019 09:00-10:00, regridded from 0.25° to 5.625° resolution using python's XESMF package.

- The WeatherBench dataset [35], which can be found at https://github.com/pangeo-data/WeatherBench. In particular the variables 10M-U-COMPONENT-OF-WIND and 10M-V-COMPONENT-OF-WIND at 5.625° resolution for the entire available period 1979/01/01 - 2018/12/31.

- The Aeolus trajectory data, which can be read using Python's SKYFIELD API from Aeolus' two-line element set given below.
  1 43600U 18066A 21153.73585495 .00031128 00000-0 12124-3 0 9990
  2 43600 96.7150 160.8035 0006915 90.4181 269.7884 15.87015039160910

Instead of using actual observations from the Aeolus satellite, we generated our own by interpolating the ERA5 data along the satellite track, whose locations are available minutely. This is so that we can compare the predictions against the ground truth to assess the performance. We use one hour of data, and hence 60 data points, to perform a spatial interpolation instead of a space-time interpolation, which is reasonable as the atmosphere hardly moves during that time period at the spatial scale of interest. Moreover, we include the weekly climatology as prior information (computed by taking the temporal average of historical global wind patterns for each of the 52 calendar weeks during the period 1979-2018 in WeatherBench), which captures general circulation patterns such as trade winds in the poles and the equator. This is equivalent to training the GP on the difference of the wind velocity from the weekly climatology.

For the kernel, we used Matérn-3/2 on the sphere and the Euclidean space (see Borovitskiy et al. [2] for the construction of Matérn kernels on the sphere), where the prior amplitude parameter was set to a fixed value (11.5 in the spherical case and 2.2 in the Euclidean case) and the length scale parameter was learnt from data. We have tried to learn the length scale initially by fitting the GP on the satellite observations and maximizing the marginal likelihood. However, this gave an unrealistically small value, likely due to the observations being too sparse: so, instead, we first trained a sparse GP on 150 randomly chosen time slices of the weatherbench historical wind reanalysis data and minimizing the Kullback–Leibler divergence of the variational distribution from the posterior (using the Adam optimizer with learning rate `1e-2`). The mean of the learnt length scales of the 150 samples was then used as the final length scale. Denoting by $k_{\mathbb{S}^2}$ the scalar Matérn-3/2 kernel on the sphere, we construct a matrix-valued kernel on the ambient space $\mathbb{R}^3$ by taking $\boldsymbol{\kappa} = k_{\mathbb{S}^2}\mathbf{I}_{3\times 3}$ as in the dynamics experiment, which is then used to construct the projected kernel with the projection given by (84). Finally, we note that when fitting the GP on the satellite observations, we use an observation error of 1.7m/s, which reflects the sum of the random and systematic error in the Aeolus satellite, as detailed by its technical specifications [12].