# OpenReview forum: "Vector-valued Gaussian Processes on Riemannian Manifolds via Gauge Independent Projected Kernels"
_NeurIPS.cc/2021/Conference — NeurIPS 2021 Poster_

### Official Review · Reviewer_mmXd · 2021-07-01

**Rating:** 8
**Confidence:** 5

**Summary:**

The paper proposes a natural extension of vector-valued GPs to data residing on Riemannian manifolds. The construction is straight-forward (replace matrix products with bilinear forms), which I consider a good thing. This also imply that it is relatively easy to apply the developed theory in practice. Toy demonstrations are provided to show that this is indeed the case. The paper is exceptionally well-written.

**Limitations And Societal Impact:**

The paper is foundational, so the societal impact is left as future work.

As to limitations, then I would have appreciated a deeper discussion of the reliance on an embedding of the manifold. This potentially prevents a great deal of applications, yet it seems that what is actually needed are operators for tangential projections, which perhaps can be found even when embeddings are not easily available. Either way, I feel that this is a significant limitation and it would be good to more explicitly acknowledge this.

**Main Review:**

The task considered in this paper is that of fitting GPs to vector fields over Riemannian manifolds. The first task is then to define the notion of GPs in this setting. The paper follows the obvious path forward and phrase this in the tangent bundle (note: 'obvious' is a positive word; I merely mean 'non-silly'), and formally verify that this is a sensible approach. Next the paper provides a scheme for extending any scalar kernel on the manifold to a vector-valued kernel. This is the main technical contribution.

The paper is a joy to read; things are clearly spelled out most of the time, and the authors have chosen a nice balance between rigor and intuition when explaining the concepts.

== Comments and questions (in arbitrary order) ==
*) In lines 117-120, one could consider mentioning that you just view matrix multiplication as the application of a bilinear operator. For readers familiar with such, a large bulk of the paper follows directly once this view is in place.

*) I found it somewhat annoying to have to go to the appendix to see a definition of a 'coframe' in Proposition 5 -- is that really needed? It would be nice of the paper was more self-contained.

*) In general, I enjoyed that the paper does not try to make things sound more complicated than they really are. This paper could easily have been written using as scary-as-possible math, and the authors did a nice job of avoiding this. One place where I found this didn't quite work out was lines 180-183 where the text reads (to me) as if it is a big challenge to apply variational approximations within the proposed setup. I'd flip this around and say that part of the elegance of the proposed approach is that this is mostly trivial.

*) I appreciate that footnote 3 makes it explicit that Nash's Theorem is about the metric tensor; all too many machine learning papers have been written assuming that Nash's theorem is concerned with the distance function.

*) I wonder if the construction of tangential projection (lines 200-211) is related to the 'projected normal distribution' (see e.g. Mardia & Jupp's book on directional statistics) ? Can I essentially think of the proposed machinery as the extension of this distribution to the infinite dimensional (i.e. stochastic process) case (and for general Riemannian manifolds) ?

*) It seems to me that the approach should also be applicable to deep kernels. These are often phrased as k'(x, y) = k(f(x), f(y)), where k is a kernel of interest, and f is a neural network. My intuition is that if f has a full-rank Jacobian everywhere, then one could still apply the machinery proposed in this paper. If so, that would be kinda neat.

*) It's a shame that the authors does not seem to plan on releasing code. I understand that for a paper of this sort, the developed code is probably messy, and not suitable as a library. But just having a Jupyter notebook that illustrated how one might do this on a sphere using GPFlow, would make it much easier for others to pick up the work. I recommend that the authors consider this.

*) It might also be good to mention other works on GPs on manifolds, e.g. "Wrapped Gaussian process regression on Riemannian manifolds", Mallasto & Aasa Feragen, CVPR 2018, comes to mind.

== After rebuttal ==
The authors give very satisfying answers to all my questions. I'm happy, and still recommend acceptance.

**Time Spent Reviewing:**

1

---

> ### Author Response · Authors · 2021-08-10
> **Response to Reviewer mmXd**
>
> We are pleased to hear that you enjoyed reading our work and we appreciate very much the detailed comments! These suggestions will certainly improve the quality of our paper and in particular.
>
> > The task considered in this paper is that of fitting GPs to vector fields over Riemannian manifolds. The first task is then to define the notion of GPs in this setting. The paper follows the obvious path forward and phrase this in the tangent bundle (note: 'obvious' is a positive word; I merely mean 'non-silly'), and formally verify that this is a sensible approach. Next the paper provides a scheme for extending any scalar kernel on the manifold to a vector-valued kernel. This is the main technical contribution.
>
> > The paper is a joy to read; things are clearly spelled out most of the time, and the authors have chosen a nice balance between rigor and intuition when explaining the concepts.
>
> Thank you for these encouraging remarks!
>
> >  In lines 117-120, one could consider mentioning that you just view matrix multiplication as the application of a bilinear operator. For readers familiar with such, a large bulk of the paper follows directly once this view is in place.
>
> We will add this clarification.
>
> > I found it somewhat annoying to have to go to the appendix to see a definition of a 'coframe' in Proposition 5 -- is that really needed? It would be nice of the paper was more self-contained.
>
> We will adopt your advice on defining "coframe" earlier in the text to make the reading experience a bit smoother.
>
> > In general, I enjoyed that the paper does not try to make things sound more complicated than they really are. This paper could easily have been written using as scary-as-possible math, and the authors did a nice job of avoiding this. One place where I found this didn't quite work out was lines 180-183 where the text reads (to me) as if it is a big challenge to apply variational approximations within the proposed setup. I'd flip this around and say that part of the elegance of the proposed approach is that this is mostly trivial.
>
> We agree with the comment regarding the simplicity of extending variational techniques to this framework, and plan to change the language to highlight that this is an advantage of this approach.
>
> > I appreciate that footnote 3 makes it explicit that Nash's Theorem is about the metric tensor; all too many machine learning papers have been written assuming that Nash's theorem is concerned with the distance function.
>
> Thanks for this remark! We agree this distinction is important!
>
> >  I wonder if the construction of tangential projection (lines 200-211) is related to the 'projected normal distribution' (see e.g. Mardia \& Jupp's book on directional statistics) ? Can I essentially think of the proposed machinery as the extension of this distribution to the infinite dimensional (i.e. stochastic process) case (and for general Riemannian manifolds) ?
>
> Thanks for bringing this construction to our attention! We believe one could view this as a sort of "tangent vector analog" of the idea behind the projected normal distribution. In our case, we project ambient vectors onto tangent planes, rather than scalar fields or densities onto the manifold directly.
>
> > It seems to me that the approach should also be applicable to deep kernels. These are often phrased as $k'(x, y) = k(f(x), f(y))$, where $k$ is a kernel of interest, and $f$ is a neural network. My intuition is that if $f$ has a full-rank Jacobian everywhere, then one could still apply the machinery proposed in this paper. If so, that would be kinda neat.
>
> Applying these kernels in a deep kernel setting could indeed be interesting! If the Jacobian of $f$ is full rank everywhere, this gives a diffeomorphism. One could possibly use an approch like this to learn the manifold that the data lives on, as long as its in the same homotopy class as the manifold you start with. One way of parametrising $f$ might be via some kind of flow vector field defined on the manifold. This could make for promising future work.
>
> > It's a shame that the authors does not seem to plan on releasing code. I understand that for a paper of this sort, the developed code is probably messy, and not suitable as a library. But just having a Jupyter notebook that illustrated how one might do this on a sphere using GPFlow, would make it much easier for others to pick up the work. I recommend that the authors consider this.
>
> We very much **do plan on releasing code**; in fact, an initial version can already be found in the supplementary materials. The code is written in a way to make implementing new manifolds and embeddings as simple as possible. We have not mentioned it in the main body to maintain our anonymity, however we will make it public in due course and link to our GitHub repository in revised versions of the text.
>
> >  It might also be good to mention other works on GPs on manifolds, e.g. "Wrapped Gaussian process regression on Riemannian manifolds", Mallasto \& Aasa Feragen, CVPR 2018, comes to mind.
>
> We agree!
> Even though this work studies (generalized) Gaussian processes $f : \mathbb{R} \to X$ which departs considerable from our setting, readers who stumble upon our paper might instead be looking for that, and we will therefore add a reference to this to the subsequent version.
>
> > The paper is foundational, so the societal impact is left as future work.
> >
> > As to limitations, then I would have appreciated a deeper discussion of the reliance on an embedding of the manifold. This potentially prevents a great deal of applications, yet it seems that what is actually needed are operators for tangential projections, which perhaps can be found even when embeddings are not easily available. Either way, I feel that this is a significant limitation and it would be good to more explicitly acknowledge this.
>
> To address the concern about the reliance on the embedding inhibiting the potential use of our work, we would like to point out that in a lot of examples that one encounters in practice, the embedding map is known either analytically, or quickly becomes available once mesh-based approximations are introduced.
> We agree that fully intrinsic techniques for defining vector-valued kernels are of great value, and believe this forms a promising avenue for future work.

---

### Official Review · Reviewer_RZVP · 2021-07-13

**Rating:** 7
**Confidence:** 4

**Summary:**

The paper suggests a construction of Covariance functions of multi-output Gaussian processes for vector fields on manifolds. The construction is independent of coordinate choices, but does depend on a Nash embedding of the manifold. In addition, the paper describes a clear mathematical framework of coordinate independent constructions of Gaussian processes. The approach is extended to variational approximations of such Gaussian processes, where the independence of the approach w.r.t. coordinate choices is again taken into consideration.

**Limitations And Societal Impact:**

None.

**Main Review:**

**edit:**
The answer to criticism (1) is very convincing. The answer to criticism (2) is as of yet rather vague. I have increased the score form 6 to 7.
---

The paper is very clear in its mathematical content (with one exception, see below), including the proofs in the appendix. In particular, both measure theory and differential geometry are correctly represented and cleverly combined. The results are correct, meaningfull and important, to construct regression models of vector fields on manifolds, e.g. the surface of the earth. Experiments are conducted, clearly described, and code is given.
In general, I think the paper deserves to be published.

However, I have two major criticisms, which I would like to see fixed. These issues decrease my rating. I am willig to increase it again after a suitable author response.
 (1) Are your constructions independent of the choise of the Nash embedding? (Probably not.) How do you justify working with these choices in a paper that is so much about naturality (=independence of choices)? Furthermore: how do you get these embeddings in practice?
 (2) Your examples (RxS^1 and S^2) are rather trivial, compared to what is possible with your approach. Can you give geometrically more complicated examples, e.g. a Klein bottle or a torus?

Some minor comments, which should be easily fixed:
 - The cited literature is coming from a background of differential geometry in neural networks. There are already paper using constructions of Gaussian processes on manifolds (in particular spheres), often including vector fields. You might want to compare your approach at least to some of those, e.g. [CL, DDH, EP, EFP, L1,L2] (Note that your approach is probably supperior to those)
 - I guess the typo spoken about on page 18 (appendix) will be fixed in the final version.
 - Line 26: the hairy ball theorem only holds for even-dimensional spheres. One could mention that the existence of smooth non-vanishing vectors fields is a rare occurence for generic manifolds.
 - Line 75: regarding "spaces with the structure of smooth manifolds", these space usually have more than one smooth structure, they only have a unique natural smooth structure constructed from the smooth structure of X.
 - You might want to give clear references to the proofs of your theorems into the appendix.
 - Line 258: "The vast majority of..." please name at least 3 that do and at least one that does not.
 - Section 4.2: Wind fields should be approximately divergence free. Could the divergence free fields on a sphere from [LH1] help?
 - Appendix: Definition 11: what are algebraic sections? I would call sections algebraic, if they are given by polynomial (=algebraic) maps. You probably mean something different. This should be cleared up.

References

  [CL] Creasey, Lang, Fast generation of isotropic Gaussian random fields on the sphere, Monte Carlo Methods and Applications 2018;

  [DDH] Dutordoir, Durrande, Hensman, Sparse Gaussian Processes with Spherical Harmonic Features, ICML 2020;

  [EP] Emery, Porcu, Simulating isotropic vector-valued Gaussian random fields on the sphere through finite harmonics approximations, Stochastic Environmental Research and Risk Assessment 2019;

  [EFP] Emery, Furrer, Porcu, A turning bands method for simulating isotropic Gaussian random fields on the sphere, Statistics & Probability Letters 2019;

  [L1] Lange-Hegermann, Algorithmic Linearly Constrained Gaussian Processes, Neurips 2018;

  [L2] Lange-Hegermann, Linearly Constrained Gaussian Processes with Boundary Conditions, AISTATS 2021.

**Time Spent Reviewing:**

4

---

> ### Author Response · Authors · 2021-08-10
> **Response to Reviewer RZVP**
>
> Thank you for giving us a very detailed feedback!
> We are delighted to hear that you appreciate our contributions!
>
> > The paper is very clear in its mathematical content (with one exception, see below), including the proofs in the appendix. In particular, both measure theory and differential geometry are correctly represented and cleverly combined. The results are correct, meaningfull and important, to construct regression models of vector fields on manifolds, e.g. the surface of the earth. Experiments are conducted, clearly described, and code is given. In general, I think the paper deserves to be published.
>
> Thank you for these encouraging remarks!
>
> > However, I have two major criticisms, which I would like to see fixed. These issues decrease my rating. I am willig to increase it again after a suitable author response. (1) Are your constructions independent of the choise of the Nash embedding? (Probably not.) How do you justify working with these choices in a paper that is so much about naturality (=independence of choices)? Furthermore: how do you get these embeddings in practice?
>
> The point made about the dependence of our method on the isometric embedding is excellent: we will add this to our description.
> It is true that our construction depends on the choice of isometric embedding: however since no expressivity is lost by our embedding approach as shown in Proposition 7, we can easily show that given a kernel $K_F$ defined via an embedding, the same kernel can be re-expressed with respect to any other embedding.
> More precisely, if $\iota_1 : X \rightarrow \mathbb{R}^{k}$ is an embedding with projection matrix $P_x$, and the kernel is
> $$
> K_F(x,x') = P_x \kappa_1(x,x') P_{x'}^T,
> $$
> where $\kappa_1$ is a matrix-valued kernel in the ambient space $\mathbb{R}^{k}$, then the same kernel can be expressed using another embedding $\iota_2 : X \rightarrow \mathbb{R}^{l}$ with projection matrix $Q_x$.
> This is done by introducing another matrix-valued kernel $\kappa_2$ in $\mathbb{R}^{l}$ defined by
> $$
> \kappa_2(x, x') = R_x^T \kappa_1(x,x') R_{x'},
> $$
> where $R_x := P^T_{x}\Gamma_x^{-1}Q_{x}$ and $\Gamma$ is the metric tensor.
> Summarizing, it is easy to express the same kernel using different embeddings, though one does still need to choose an embedding initially to set a kernel.
>
> This means that our construction, while not completely intrinsic, is at least sufficiently flexible to give practitioners a wide class of kernels that can be constructed in a manner tailored to the problem at hand.
> Finding a natural choice of embedding would provide an interesting direction to pursue for further study, and might depend on the specific manifold at hand, or on certain properties one might wish the kernel to have---for example, one might seek a projected kernel arising from some definition of a vector heat (diffusion) kernel on a given manifold.
> These cases provide fruitful opportunities for future work.
>
> > (2) Your examples ($R \times S^1$ and $S^2$) are rather trivial, compared to what is possible with your approach. Can you give geometrically more complicated examples, e.g. a Klein bottle or a torus?
>
> To construct embeddings for more complicated geometrical objects, most manifolds encountered in practice split into two main cases.
>
>  * Case (i) involves manifolds for which explicit expressions for embedding maps are available, such as the $n$-sphere, $n$-torus, Klein bottle, compact semi-simple Lie groups, and many spaces of interest in physics.
> We intend to expand the number of examples in the final paper and code base, to include a number of these more complex examples.
>  * Case (ii) involves manifolds represented numerically using a mesh, such as the famous Stanford bunny rabbit manifold widely-used as a benchmark example in computer graphics and other areas. For this case, the mesh itself provides an embedding of a 2D manifold into 3D space. Here, we can take advantage of prior work for computing kernels on meshes to construct vector valued kernels on meshes via our construction.
>
> > The cited literature is coming from a background of differential geometry in neural networks. There are already paper using constructions of Gaussian processes on manifolds (in particular spheres), often including vector fields. You might want to compare your approach at least to some of those, e.g. [CL, DDH, EP, EFP, L1,L2] (Note that your approach is probably supperior to those)
>
> Thank you for making us aware of these works! We agree that they are relevant to our work and will include them in the citations.
>
> > I guess the typo spoken about on page 18 (appendix) will be fixed in the final version.
>
> Thank you - we will fix this in the final version.
>
> > Line 26: the hairy ball theorem only holds for even-dimensional spheres. One could mention that the existence of smooth non-vanishing vectors fields is a rare occurence for generic manifolds.
>
> Thank you for pointing this out - we will add this clarification.
>
> > Line 75: regarding "spaces with the structure of smooth manifolds", these space usually have more than one smooth structure, they only have a unique natural smooth structure constructed from the smooth structure of X.
>
> Thanks again for pointing this out. We will change the line to "spaces with the structure of smooth manifolds induced by the smooth structure of $X$".
>
> > You might want to give clear references to the proofs of your theorems into the appendix.
>
> We will add specific references to each proof in the next version of the paper.
>
> > Line 258: "The vast majority of..." please name at least 3 that do and at least one that does not.
>
> Every single inducing point construction in the papers cited in the Line 258 makes use of matrix-vector expressions in constructing inducing points approximations. On the other hand, we prefer not claiming that all constructions do so due to the vastness of the literature.
>
> > Section 4.2: Wind fields should be approximately divergence free. Could the divergence free fields on a sphere from [LH1] help?
>
> Horizontal wind fields at sufficiently high altitudes are indeed approximately divergence-free, in which case it would be possible to apply techniques for divergence-free vector fields on the sphere such as [LH1]. However, this is not the case in the atmospheric boundary layer, which is the regime of up to around 1000m from sea level. In this setting, imposing a divergence-free requirement on the horizontal wind field may be too restrictive.
>
> > Appendix: Definition 11: what are algebraic sections? I would call sections algebraic, if they are given by polynomial (=algebraic) maps. You probably mean something different. This should be cleared up.
>
> We used the term "algebraic sections" to explicitly distinguish them from smooth sections: the former object does not require smoothness. If $s : X \rightarrow TX$ and $\pi : TX \rightarrow X$ is the canonical projection, the "algebraic" here refers to the fact that $s$ satisfies the algebraic condition $\pi \circ s = id_X$ (that is, it is a section in the category-theoretical sense) - this does not refer to $s$ being defined by polynomials. We will clarify this in a short remark.

---

### Official Review · Reviewer_aB3f · 2021-07-17

**Rating:** 5
**Confidence:** 4

**Summary:**

This paper proposed a way to define vector-valued Gaussian processes on Riemannian manifolds. A computational framework is provided to facilitate statistical inference using these models. The main contribution is an explicit methodology for characterizing vector-valued Gaussian processes through extrinsic kernels satisfying a gauge constraint, thus bypassing abstract mechanisms often employed to define stochastic processes on Riemannian manifolds. Two examples are provided to illustrate the power of the proposed models.

**Limitations And Societal Impact:**

While the authors emphasized the value of the theoretical constructions in this paper, I found the computational ingredients and illustrated examples much more valuable since the theoretical construction does not appear novel given the extensive existing literature on stochastic differential geometry. More specifically:

- It is not immediately clear to me why the authors indicated in the introduction that defining vector-valued Gaussian process on Riemannian manifold is harder than defining scalar-valued Gaussian processes on manifolds or manifold valued Gaussian processes. Manifold-valued diffusion processes are well-understood for at least several decades [1, Sec. III]; the extrinsic characterization and projection approach can be traced back to at least [2, Ch. 4]; many recent work in statistical learning have addressed similar and more general themes covered in this submission, e.g. [3], [4], [5], [6]. The equivariant viewpoint played a central role in [4], and the projection of extrinsic processes to tangent spaces is known to [1], [2] and more generally the stochastic geometry community. The gauge-equivariance of the kernel and the variational approximations can be derived from these results without too much extra technicality. It is true that this submission phrased many of these ideas in a different context, but the novelty is limited due to the vast existing literature covering this topic, and relation of proposed work with these previous contributions are not fully elaborated.

[1] Émery, M. (2012). Stochastic calculus in manifolds. Springer Science & Business Media.

[2] Stroock, D. W. (2000). An introduction to the analysis of paths on a Riemannian manifold (No. 74). American Mathematical Soc.

[3] Mallasto, A., & Feragen, A. (2018). Wrapped Gaussian process regression on Riemannian manifolds. In Proceedings of the IEEE Conference on Computer Vision and Pattern Recognition (pp. 5580-5588).

[4] Lin, L., Mu, N., Cheung, P., & Dunson, D. (2019). Extrinsic Gaussian processes for regression and classification on manifolds. Bayesian Analysis, 14(3), 887-906.

[5] Mallasto, A., Hauberg, S., & Feragen, A. (2019, April). Probabilistic Riemannian submanifold learning with wrapped Gaussian process latent variable models. In The 22nd International Conference on Artificial Intelligence and Statistics (pp. 2368-2377). PMLR.

[6] Guhaniyogi, R., & Dunson, D. B. (2016). Compressed gaussian process for manifold regression. The Journal of Machine Learning Research, 17(1), 2472-2497.

- I believe the proposed computational methodology will have significant impact, but more examples will need to be provided to illustrate the computational advantage. Approximation errors incurred in the variational approximation and their dependence on the manifold geometry may need further elaboration as well.

**Main Review:**

Originality: Extending previous work on Gaussian processes on Riemannian manifolds to vector fields is an interesting and valuable contribution. It is not clear why previous work can not be more straightforwardly generalized to this setting, and a lot of literature on stochastic processes on Riemannian manifolds are not mentioned.

Quality: This submission is technically sound and complete.

Clarity: This submission is reasonably clearly written, well organized, and easy to follow.

Significance: The results presented in this paper have great practical value. It can be imagined that this work can potentially lead to future work along the same lines but larger societal impacts, as the authors suggested in the conclusion section.

**Time Spent Reviewing:**

8 hours

---

> ### Author Response · Authors · 2021-08-10
> **Response to Reviewer aB3f**
>
> Firstly, thank you very much for the review!
> We are deeply grateful for your time reviewing our work and providing detailed, constructive feedback, and for describing the work as having "great practical value"!
> The feedback is very helpful for us and provides important insights on how we can further improve our work.
> Below, we would like to address the two key issues raised, namely concerns regarding (1) novelty and (2) experiments to illustrate the computational advantage of our approach.
>
> > Originality: Extending previous work on Gaussian processes on Riemannian manifolds to vector fields is an interesting and valuable contribution. It is not clear why previous work can not be more straightforwardly generalized to this setting, and a lot of literature on stochastic processes on Riemannian manifolds are not mentioned.
>
> The vector field setting ($f: X \to TX$) is more challenging than it may look at first - from a technical perspective, it is significantly different from the scalar-valued setting ($f: X \to \mathbb{R}$).
>
> To illustrate this with an example, consider that in our setting, it is at first unclear what the appropriate notion of a "covariance kernel" should be.
> The standard Euclidean notion of a matrix-valued kernel $k : X \times X \to \mathbb{R}^{d\times d}$ clearly does not transfer to the vector-field setting without further considerations, due to the dependence of the matrix output on the choice of tangent bases.
> Compare this to the scalar-valued case, where the notion of a kernel generalizes immediately and no such issue arises.
> We provide a suitable analog.
>
> Our key contribution is transferring an appropriate differential-geometric language from the mathematical literature into this setting, understanding what the correct analogues of standard Gaussian process notions are for vector fields, and providing concrete expressions which enable practitioners to implement such algorithms in modern systems such as TensorFlow, PyTorch, or JAX.
>
>
> >Quality: This submission is technically sound and complete.
> >
> > Clarity: This submission is reasonably clearly written, well organized, and easy to follow.
> >
> > Significance: The results presented in this paper have great practical value. It can be imagined that this work can potentially lead to future work along the same lines but larger societal impacts, as the authors suggested in the conclusion section.
>
> Thank you for these encouraging remarks!
>
> > While the authors emphasized the value of the theoretical constructions in this paper, I found the computational ingredients and illustrated examples much more valuable since the theoretical construction does not appear novel given the extensive existing literature on stochastic differential geometry. More specifically:
> >
> > It is not immediately clear to me why the authors indicated in the introduction that defining vector-valued Gaussian process on Riemannian manifold is harder than defining scalar-valued Gaussian processes on manifolds or manifold valued Gaussian processes. Manifold-valued diffusion processes are well-understood for at least several decades [1, Sec. III]; the extrinsic characterization and projection approach can be traced back to at least [2, Ch. 4]; many recent work in statistical learning have addressed similar and more general themes covered in this submission, e.g. [3], [4], [5], [6]. The equivariant viewpoint played a central role in [4], and the projection of extrinsic processes to tangent spaces is known to [1], [2] and more generally the stochastic geometry community. The gauge-equivariance of the kernel and the variational approximations can be derived from these results without too much extra technicality. It is true that this submission phrased many of these ideas in a different context, but the novelty is limited due to the vast existing literature covering this topic, and relation of proposed work with these previous contributions are not fully elaborated.
>
> The references provided use similar technical tools to solve **different** problems.
> Specifically,
>
>  * References [1], [2], [3] and [5] deal primarily with the manifold-valued case ($f : \mathbb{R} \to X)$, which involves different considerations from our setting. In particular, references [3] and [5] consider Gaussian processes that take values in manifolds, whereas [1] and [2] are concerned with constructing manifold-valued Itō processes, which largely constitute a different model class to Gaussian processes. We agree that the embedding approach in [2] is similar in spirit to our ideas (albeit in a different setting) and therefore should certainly be mentioned in §3.1 where we introduce projected kernels - thank you for making us aware of this work!
>  * The reference [4] is concerned with constructing scalar-valued GPs ($f : X \to \mathbb{R}$), utilising an embedding in Euclidean space to explicitly construct scalar kernels on manifolds. In contrast, we use intrinsic scalar kernels, and use the embeddings to construct smooth basis fields. [4] Does not consider vector fields, and therefore deals with a different set of differential-geometric subtleties.
>  * The reference [6] studies random compression matrices for dimensionality reduction, using the word "manifold" to mean "non-linear subspace of $\mathbb{R}^d$", which departs considerably from our setting.
>
> We will add references to these settings to the next version. This way, readers who are looking for them but stumble upon our work can find these papers more easily.
>
> While we agree that the technical machinery used in our work---for instance, (random) vector fields on manifolds, and techniques based on projections---is classical and well-understood both in the mathematics and theoretical physics communities, we believe that the application of this machinery to the Gaussian process setting in machine learning is novel and worthy of study in its own right.
>
> In an area where much of the prior work concerns only the sphere and other special cases, our work lays the foundation for applying the Gaussian process and statistical decision-making toolbox to novel problems in the physical sciences and other areas.
>
> > I believe the proposed computational methodology will have significant impact, but more examples will need to be provided to illustrate the computational advantage.
>
> Within machine learning, the setting we study is sufficiently unexplored that no general alternative Gaussian process method is available for the experiments we conduct.
> Our goal here is to improve generality, rather than performance, providing usable techniques for the vector field setting.
> Since by Proposition 7, all vector-valued kernels arise using the projected construction, our method includes previous vector-valued kernels on specific spaces such as the sphere as a special case.
>
> > Approximation errors incurred in the variational approximation and their dependence on the manifold geometry may need further elaboration as well.
>
> We agree that studying the approximation error of variational approaches is a worthy and interesting topic!
> These properties will depend on the details of the variational approximation considered: if there are sufficiently many inducing points, it is known that many forms of inducing-point-based variational approximations recover the true posterior---by the intrinsic nature of our construction, these properties extend directly to our setting.
> Outside of this regime, approximation error in variational methods is currently an open research topic even in the Euclidean case where geometric issues do not appear.
> Therefore, we prefer to leave this important topic to subsequent work.

---

### Official Review · Reviewer_UAbr · 2021-07-19

**Rating:** 9
**Confidence:** 4

**Summary:**

The authors propose a class of vector Gaussian processes (GP) on a class of Riemannian manifolds.  An important feature of the proposed GP is its ability to model vector fields on the Riemannian manifold. This is a broad application in physical sciences. One of the main novelties of the work is the development of gauge-equivariant kernels operators. Variation inference algorithms are extended for posterior inference.  Numerical examples demonstrate the practical performance of the proposed GP.

**Limitations And Societal Impact:**

Comparisons with deep GP are missing.

**Main Review:**

Novel contributions are as follows:
* The authors propose a class of vector Gaussian processes (GP) on a class of Riemannian manifolds.
* The authors successfully apply techniques from differential geometry for flexible GP modeling.
* The authors develop gauge-equivariant kernels operators.
* Outline the approach for posterior inference using variational Bayes.

**Time Spent Reviewing:**

2

---

> ### Author Response · Authors · 2021-08-10
> **Response to Reviewer UAbr**
>
> Thank you for taking the time to review our work!
> We are very pleased to hear that you appreciate our contributions and are encouraged by your comments.
>
> > Novel contributions are as follows:
> >
> >The authors propose a class of vector Gaussian processes (GP) on a class of Riemannian manifolds.
> > The authors successfully apply techniques from differential geometry for flexible GP modeling.
> > The authors develop gauge-equivariant kernels operators.
> > Outline the approach for posterior inference using variational Bayes.
>
> Thank you for these encouraging remarks!
>
> > Comparisons with deep GP are missing.
>
> Using deep Gaussian processes to model vector fields on manifolds requires significant further technical development.
>
> The issue here is that one needs to think carefully about how to define an appropriate latent state space for such a model: one could consider for instance a manifold-valued latent state space where intermediate layers map the tangent bundle to itself.
> Even this is technically non-trivial, because it requires one to understand what ought to be meant by Gaussianity in this setting, which for a map $f : TX \to TX$ will be different than for $f : X \to TX$ (our setting) or for $f : \mathbb{R} \to X$ or $f : X \to \mathbb{R}$ (previous works).
> Most non-Euclidean alternatives present similar challenges, which are sufficient in scope that a full paper could be written about those issues.
>
> We hope that our work provides a technical foundation on top of which some of these ideas could be developed.

---

### Decision · Program_Chairs · 2021-09-27

**Decision:**

Accept (Poster)

**Comment:**

The majority of the reviewers argued rather strongly for accepting this paper. One reviewer saw the paper as more borderline, but the concerns were addressed in the rebuttal. This paper is to most parts well written and, as one of the reviewers say, a joy to read. However, the reviewers have all made a number of remarks for improving the paper, and I recommend that you take care in incorporating them in the camera-ready.